# Human class B1 GPCR modulation by plasma membrane lipids
Kin W. Chao[1,2,4], Linda Wong[3], Affiong I. Oqua[3], Jas Kalayan [2], Yusman Manchanda[3], James Gebbie-Rayet [2], George Hedger [1]✉, Alejandra Tomas [3]✉ & Sarah L. Rouse [1]✉

The class B1 G protein-coupled receptor (GPCR) subfamily is a class of receptors known for their regulatory roles in metabolism and neuronal activity and as important drug targets. Lipids play key functional roles in modulation of GPCR signalling, yet our understanding of the molecular level detail of specific lipid interactions with class B1 GPCRs remains limited. Here we present coarse-grained molecular dynamics (MD) simulations of the active and inactive states of 15 human class B1 family members and use aiida-gromacs to capture full provenance for the set-up of simulations in complex plasma membranes. Receptors exhibit state-dependent lipid interactions with the regulatory sterol cholesterol and phospholipid phosphatidylinositol-3,4-bisphosphate (PIP$_2$) at defined locations on the receptor surface. Global analysis of trends across the subfamily reveals conserved patterns of lipid interaction dynamics. The glycosphingolipid GM3 exerts a modulatory influence on the dynamics of class B1 extracellular domains in both simulations and in vitro time-resolved FRET assays.

Class B1 G protein-coupled receptors (GPCRs) are a subfamily of 15 membrane proteins, which are grouped into 5 further subfamilies: Glucagon-like, PTH-like, CT-like, CRF-like and PACAP/VIP-like[1] (Fig. 1A). These receptors are involved in regulating critical biological processes such as metabolism and neuronal activity and are important drug targets for multiple diseases[2]. The substrates of class B1 GPCRs are polypeptide hormones, and these receptors are typically characterised by a having a large extracellular domain (ECD) believed to be important in the binding affinity of peptide agonist and signal transduction[3]. Within the membrane region their structure follows the canonical GPCR seven transmembrane helix bundle. Upon activation, the principal conformational change is the outward movement of the cytoplasmic portion of transmembrane (TM) helix TM6. This exposes a shallow groove with which select G proteins may associate, leading to nucleotide exchange and propagation of signalling to downstream cascadikes[4,5].

The plasma membrane environment in which class B1 receptors reside is a key modulator of membrane protein function[6,7]. Several lipid species are reported to have modulatory effects on GPCRs, including cholesterol, phosphatidylinositol-4,5-bisphosphate [(PI4,5)P$_2$], and the glycosphingolipid GM3[8–13]. These modulatory lipids can influence multiple aspects of GPCR function, including the conformational landscape, G protein and arrestin coupling, and oligomerisation state[14–16]. Such modulation can be brought about by altering the biophysical properties

of the local membrane environment, as well as by specific binding of modulatory lipids to allosteric sites on the GPCR[16,17]. Within the class B1 subfamily, cholesterol has been shown to modulate GLP1R signalling responses and biased agonism[18]. The presence of cholesterol is also required for GLP1R internalisation, optimal receptor clustering and cAMP responses[19]. However, the molecular level interactions and dynamics which underlie these functional effects and their conservation across the wider subfamily remain obscure.

Recent technological development of cryo-electron microscopy (cryo-EM) approaches[20] for GPCR structure determination has led to the elucidation of the active state conformations of all 15 class B1 GPCRs[21]. However, interpretation of lipid interactions is currently limited due to the use of detergent micelles and non-native lipid compositions within nanodiscs. Likewise, the inactive states of many class B1 GPCRs remain unresolved. In the present study, we leverage advances in simulation methodology[22], experimental lipidomics[23], and GPU computing[24], to reunite experimentally determined structures with their complex plasma membrane environment using large-scale physics-based MD simulations across the entire subfamily. We further leverage recent advances in protein structure prediction models[25,26] to interrogate the lipid interactions of the predicted inactive states. During the writing of this manuscript, a new AI foundation model, Chai-1, emerged, which enables the prediction of small molecules in complex with a given protein with AF3-like accuracy[27]. We applied Chai-1 to

[1]Department of Life Sciences, Sir Ernst Chain Building, Imperial College London, London, UK. [2]Scientific Computing Department, Science and Technology Facilities Council, Daresbury Laboratory, Warrington, UK. [3]Department of Metabolism, Digestion and Reproduction,Section of Cell Biology and Functional Genomics, Imperial College London, London, UK. [4]Present address: School of Chemistry and Chemical Engineering, University of Southampton, Southampton, UK. ✉e-mail: g.hedger@imperial.ac.uk; a.tomas-catala@imperial.ac.uk; s.rouse@imperial.ac.uk

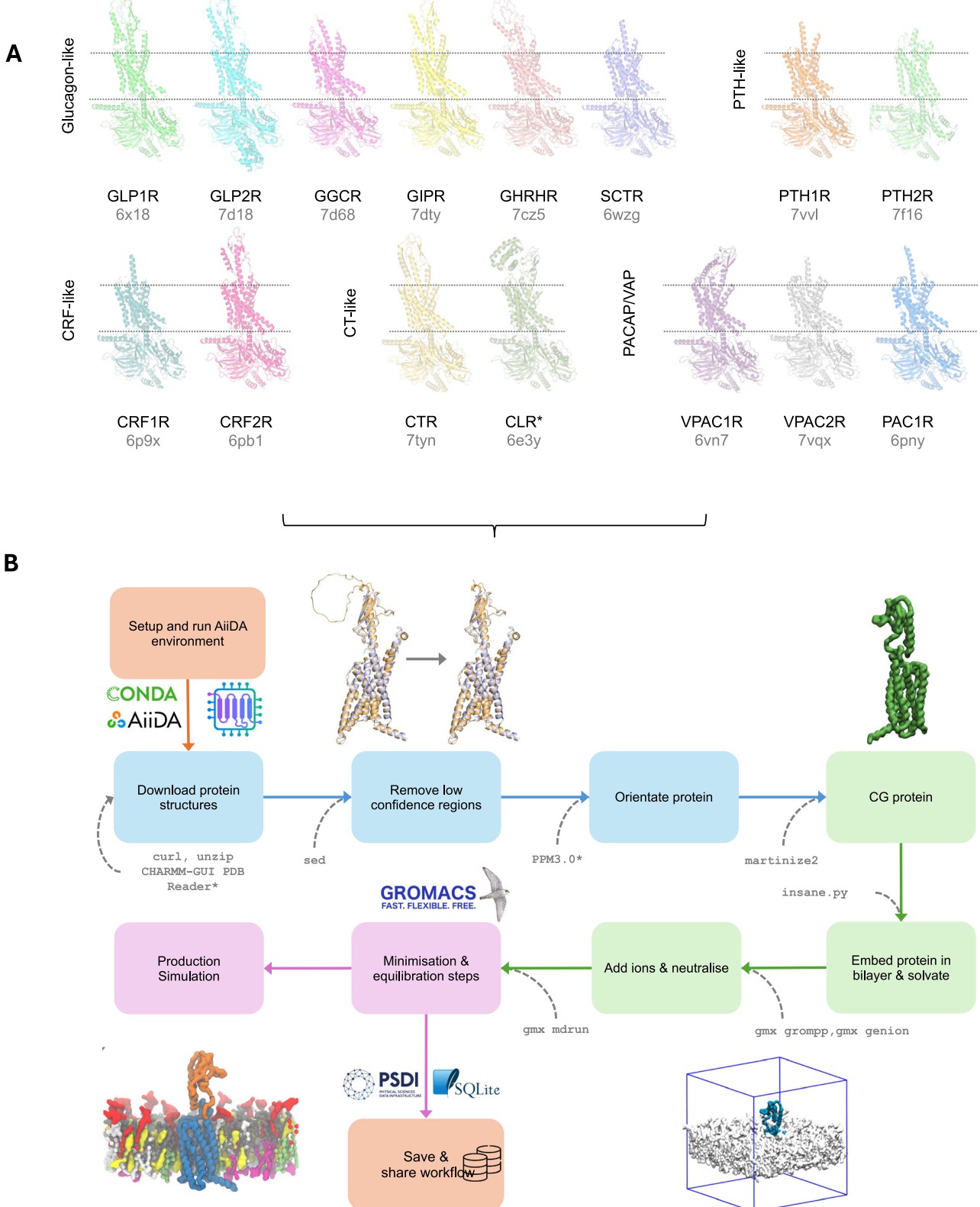

**Fig. 1 | Modelling and simulation workflow for class B1 GPCRs. A** Structures of the 15 class B1 GPCRs in the active state used here with associated PDB accession codes. For CLR, the structure of CGRP used as the template in GPCRdb is shown. The approximate position in the membrane is shown by dotted lines. **B** Schematic of the B1 GPCR simulation workflow steps captured using aiida-gromacs. Orange boxes represent steps involved in enabling provenance capture and sharing, blue boxes represent steps for processing the protein structure, green boxes represent parametrisation and building of the molecular system, and purple boxes represent MD simulation steps. The CLI tools used in each workflow step are shown in grey text. CHARMM-GUI PDB Reader[70] was used to add missing atoms, and PPM3.0[74] was used to predict membrane orientations; these steps are not yet captured by AiiDA-gromacs.

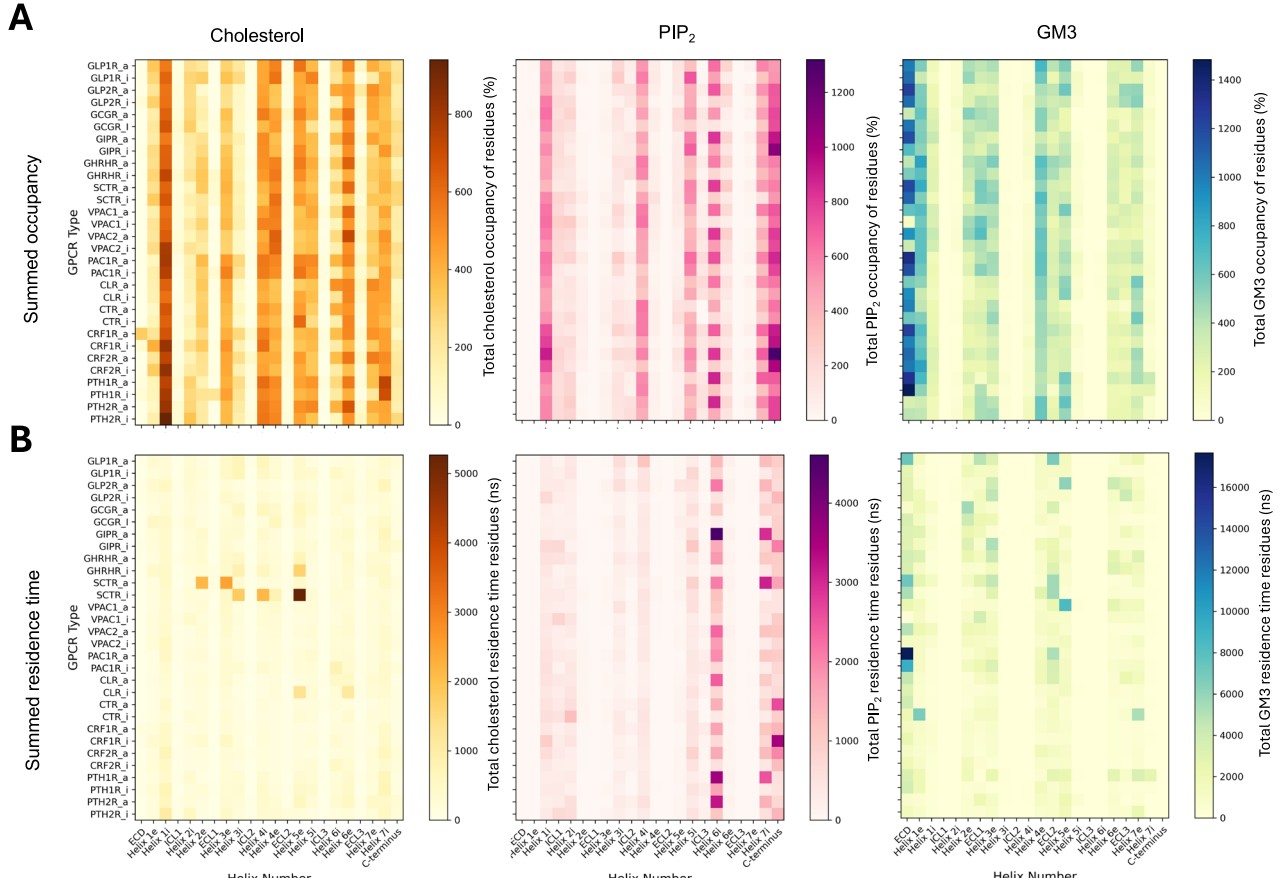

**Fig. 2 | Global analysis of class B1 GPCR interactions with the regulatory lipids cholesterol, PIP$_2$ and GM3. Lipid occupancy.** Panel **A** and lipid residence time **B** is shown for each class B1 GPCR as a function of ECD, helix, and loop number. Helix 1e denotes the extracellular half of Helix 1, and Helix 1i the intracellular half. Active states are denoted by _a, and inactive by _i. Per residue lipid occupancies were computed over 3 x 10 μs of production run simulation time using PyLipID[30] and summed over the constituent residues of each region. The residue ranges defining each region correspond to those from GPCRdb[68].

predict GPCR-cholesterol, and GPCR-PI(4,5)P$_2$ complexes, and integrated our findings with the results of physics-based MD simulations. Overall, we identify a conserved pattern of PIP$_2$ binding at the functionally important TM6/7-H8 interface, a number of deep cholesterol binding sites, and GM3 modulation of ECD dynamics, supported by in vitro time resolved FRET assays.

## Results

We simulated each receptor in a model plasma membrane environment, using the Martini 3 coarse grained force field. We generated active and inactive state models of each class B1 receptor. Available experimental structures of each receptor are shown in Fig. 1A and summarised in SI Table 1. Each simulation was performed in triplicate for 10 μs, totalling approximately 1.0 ms simulation time.

The vast array of methods and tools used to setup initial atomic configurations and compositions has resulted in complications in capturing protocols used in biomolecular simulations[28]. With scientific research methodologies focused on *why* and *what* was done to produce simulations, the *how* is often missing, which causes difficulties in reproducing and extending research. Our study uses over 30 different simulation setups with over 60 independent trajectories to analyse, necessitating novel approaches to capture full provenance. Here we present the first application of aiida-gromacs to capture all the metadata associated with this study to allow complete MD provenance including the simulation workflow and analysis (Fig. 1B). The AiiDA python framework[29] extensively used in first-principle calculations in computational chemistry, provides a powerful framework for describing the flow of data and enabling the reproduction of complex scientific workflows. The aiida-gromacs plugin (https://aiida-gromacs.

readthedocs.io) extends AiiDA to enable the capture and description of biomolecular MD simulations using the GROMACS package. By providing all the metadata as well as the production trajectories and PyLipID analyses, we aim to facilitate greater accessibility to our simulations, towards enabling reproducibility and machine learning approaches.

### Broad trends in cholesterol, GM3 and PIP$_2$ interactions for class B1 GPCRs

Analysis of total lipid residence time, binding site occupancy (Fig. 2), and membrane contact properties (SI Fig. 1) in active *versus* inactive states across all Class B1 receptors revealed a number of trends for the known regulatory lipids cholesterol, GM3, and PIP$_2$. Calculations for the non-regulatory lipid POPC, are also shown in SI Fig. 1. We map the occupancy and binding site profile for the active and inactive states cholesterol, GM3, and PIP$_2$ for each receptor onto their respective sequences using the PyLipID[30] package (SI Fig. 2–4). To assess if the combined 30 μs sampling time per receptor state is sufficient we used SCTR as a test case and performed longer timeframe (3 x 100 μs) simulations of SCTR in each state, using different initial membrane configurations (SI Fig. 5). These properties, as well as 3D volumetric density maps (SI Fig. 6) are comparable for the 10 and 100 μs timeframes. Further analysis includes an evaluation of lipid diffusion properties across these timescales (SI Fig. 7) as well as estimated errors for calculated cholesterol occupancies per residue for 10 μs and 100 μs (SI Fig. 8). These analyses indicate that three independent replicas each of 10 μs duration is sufficient to capture the lipid binding properties of SCTR.

All receptors show interactions with cholesterol, PIP$_2$ and GM3, with each lipid exhibiting distinct interaction fingerprints with defined regions of the receptors. Within the inner leaflet of the membrane, the most

pronounced changes between active and inactive states for cholesterol and $PIP_2$ contacts occur in the functionally critical TM6 region, which undergoes an outward movement upon activation (SI Fig. 1–4). In contrast, GM3 interactions show no apparent patterns between inactive and active states, with ECDs of both states contacting GM3 to differing degrees. Notable binding sites based on occupancy are observed for cholesterol (TM1), $PIP_2$ (TM6/7-H8), and GM3 (ECD). The overall residence times are observed to be in the order GM3 > $PIP_2$ > cholesterol, whilst lipid occupancy is GM3 ~ $PIP_2$ > cholesterol. For the non-regulatory lipid POPC, only PAC1R in the active state displayed high residence time binding (SI Fig. 1).

As well as capturing the overall patterns of lipid interactions across the B1 receptors, these overview plots allow us to highlight multiple outliers to characterise further. We explore these properties in further detail in the following sections.

### Putative novel cholesterol binding sites for class B1 GPCRs

We analysed the top predicted cholesterol binding sites in terms of residence time and occupancy (SI Fig. 9, 10). Several class B1 receptor structures have been resolved in the presence of cholesterol hemisuccinate, with cholesterol modelled into the map density[31–37] (Fig. 3A). Across these, a single cholesterol binding site is conserved at TM4, with the conserved residue $W^{4.50}$, using the Wootten numbering scheme[38], proposed to mediate this interaction[36,37]. A plot of the cholesterol occupancy mapped onto the aligned sequences indicates that the $W^{4.50}$ interaction is conserved during our simulations (Fig. 3B, SI Fig. 2). An analysis of all cholesterol headgroup positions mapped to the receptor TM bundle shows that these poses are sampled during our simulations (Fig. 3C). However, the top cholesterol sites defined in terms of occupancy and residence time corresponded to different receptor binding sites (SI Fig. 9,10).

Approximately 40% of the top cgMD sites across all active and inactive receptor states correspond to the polar hydroxyl group of cholesterol binding in the centre of the membrane, with one or more such sites existing for every receptor (SI Fig. 9). This "deep membrane" binding mode of cholesterol[39], with cholesterol perpendicular to the membrane and its hydroxyl group buried between two TM, has been previously observed in several GPCR simulation studies[39–42] but to our knowledge has not been captured in X-ray crystallography or cryo-EM structures of any GPCR to date, although NMR approaches have determined different affinity cholesterol binding sites[43]. We also observe deep membrane binding modes where the sterol groups of the cholesterol are packed against the TM bundle (SI Fig. 9). Defining contacts as transient (<20 ns), short (20–100 ns), medium (100–300 ns) or long-lived (>300 ns)[44] we observe only two instances of long-lived cholesterol binding, both for SCTR, which we analyse further below. Across the B1 family, for deep membrane sites the transient:short:medium ratio is 0:59:39, and for the canonical (bulk membrane orientation) binding mode the transient:short:medium ratio is 16:68:14. Therefore the deep membrane binding modes shift towards longer timeframe binding, whilst the 'normal' binding modes include high occupancy, transient binding of cholesterol,

We used a recently released multi-modal foundation model AI, Chai-1[27], to predict cholesterol-receptor complexes across all 15 class proteins (SI Fig. 11, 12). The Chai-1 webserver enables lipids to be inputted as SMILES strings, alongside the protein sequence. Chai-1 predicted the conserved TM4 site with most other predicted sites corresponding to those observed in the cryo-EM structures (Fig. 3A, Supplementary Movie 1). No cholesterol deep membrane binding modes were observed. This likely reflects the model training, which is in part based on PDB X-ray and cryo-EM structures, which may be considered to contain a 'biased sample' of tight binding lipids[45]. That is, those which bind tightly enough to survive the purification, detergents, and non-native temperatures[46] used during structure determination. The dynamic 'greasy patch' binding of cholesterol molecules we observe in MD simulations, and particularly deep membrane cholesterol, is likely to be challenging to resolve by these methods.

A comparison of the cholesterol site residence times across the class B1 GPCRs highlights a clear outlier in the case of the secretin receptor (SCTR,

Fig. 1A). The top ranked replica of the active state contained a binding site with a residence time of 692 ns compared to a range of 50–255 ns for other active state receptors. The longest residence time in the inactive the state was 1854 ns (compared to 72–170 ns for inactive states of other receptors). Whilst we cannot readily compute affinity from long binding events observed in single replicates, we assessed these sites further. When compared to the active state cryo-EM structure, we observe a shift in the position of His211 in the ECL1 loop during the simulation, which forms a contact with the cholesterol hydroxyl group in the active state (Fig. 3D). For the inactive state, the cholesterol adopts a deep membrane binding mode and buries into the gap between TM4 and TM5 (Fig. 3E) The key residues responsible for mediating the contacts in both sites are shown in Fig. 3E, D. The Chai-1 prediction does not resolve this pose (SI Fig. 13), likely due to the model plasma membrane environment being required for the conformational change of His211.

To further evaluate the deep membrane cholesterol binding mode of SCTR we used CG2AT2[47] to convert cgMD frames corresponding to the active and inactive long residence time bound cholesterol sites (SI Table 2) as the starting point for 500 ns atomistic simulations. It is known that cholesterol accesses the membrane core more frequently in coarse-grained simulations compared to atomistic simulations[48,49]. We observe this in our simulations, where cholesterol presence in the membrane core is greatly reduced in atomistic simulations, but non-zero (SI Fig. 14). For the active state containing the long-binding time deep membrane bound cholesterol, we observed dissociation of these molecules during the equilibration and minimisation steps. For the third replicate, we observed hydrogen bonding disruption within the first 10 ns, and complete dissociation of the molecule within the first 200 ns. In this replicate, a second cholesterol molecule accessed the binding site from bulk membrane at ~420 ns and remained bound for the duration of the simulation, indicating that the site identified via cgMD simulations (Fig. 3F) may also be accessed by cholesterol at atomistic resolution. Comparative lipid occupancy heatmaps for cholesterol, $PIP_2$ and GM3 for SCTR at cgMD and atomistic resolution are shown in SI Fig. 15.

### Conserved $PIP_2$ interactions at TM6/7 and H8

The inner leaflet lipid $PIP_2$ interacted with all class B1 receptors with maximum residence times in the range 955–6455 ns (active) and 216 to 1273 ns (inactive) (Fig. 4A, Supplementary Movie 1). These interactions were concentrated at the intracellular tips of TM6/7, and either side of H8 (SI Fig. 3). For all receptors, $PIP_2$ interactions at the TM6/7 site showed enhanced stability in simulations of the active state compared to the inactive state (SI Fig. 3, 9). This site was also predicted by Chai-1, along with a second site between TM2/4 (Fig. 4A). Analysis of the degree of direct contact that each residue formed with $PIP_2$ during simulations exhibited a similar trend, with increased levels of per residue contact in the active state (SI Fig. 3). We analysed the top active and inactive state site for GIPR which showed one of the highest residence times for $PIP_2$ in the active state in our simulations (Fig. 4C, D). Particularly high levels of interaction were seen between basic residues and the anionic 3' and 4' phosphoryl moieties of the $PIP_2$ headgroup. These residues are highly conserved across class B1 GPCRs (Fig. 4B). Binding of $PIP_2$ lipids was also seen on the opposing side of H8, adjacent to TM1. Binding at this location is similarly driven by contact with basic residues in H8 and the adjacent TM1 (Fig. 4D).

We analysed $PIP_2$ binding of GIPR at atomic resolution by converting the final frames of each cgMD simulation replicates and performed 500 ns atMD simulations. In these, the high residence time binding sites observed in cgMD were maintained throughout the simulations for both the active and inactive state, with some sidechain reorientations at atomistic resolution, with the Lys397 interaction reduced on this timeframe in the active state (Fig. 4C, D). In each case the key interactions involve charged Arg/Lys residues. SI Fig. 15 shows occupancy heatmaps for GIPR at cgMD and atomistic resolution.

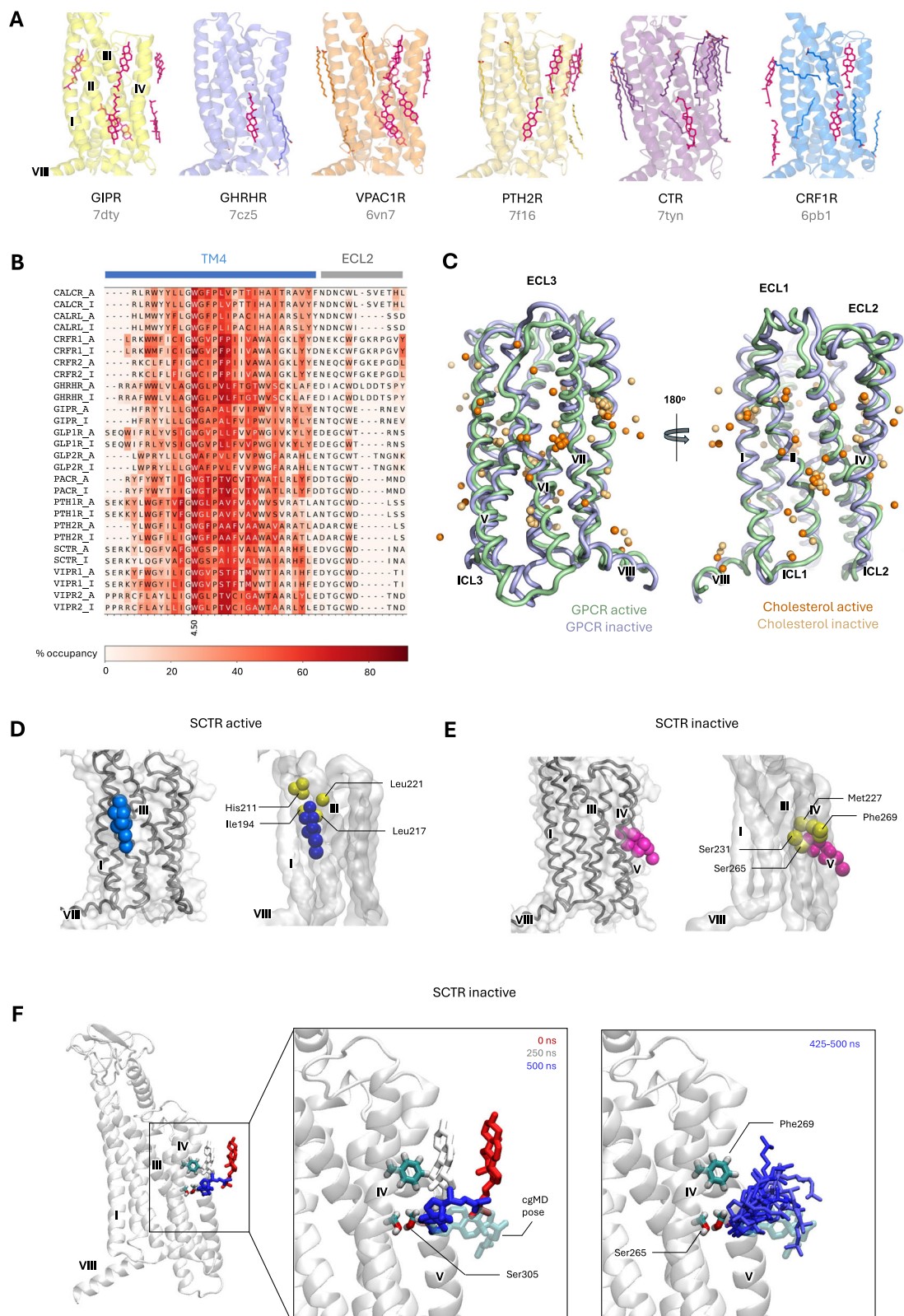

**Fig. 3 | Cholesterol interactions with class B1 GPCRs. A** Published experimental structures of class B1 receptors containing resolved cholesterol molecules. Structures of the TM regions are aligned, with helix labels shown for GIPR. Cholesterol and cholesterol analogue molecules are shown in magenta stick representation. **B** Cholesterol occupancy from cgMD simulations mapped onto sequence alignment across all 15 receptors. **C** Positions of cholesterol molecules from simulations mapped onto the CTR receptor. Active (green) and inactive (blue) states of the CTR are shown as a backbone trace. Spheres represent hydroxyl headgroups of cholesterol for the top ranked interaction sites extracted from simulations of active (dark orange) and inactive (light orange) states of all 15 class B1 GPCRs. **D** The highest residence time cholesterol site from cgMD simulations for SCTR in the active state, with residues within 6 Å of the cholesterol hydroxyl group shown as yellow spheres. **E** As (**D**) for the SCTR inactive state. **F** Cholesterol spontaneously entering the binding pocket during atMD simulation, with positions at given simulation frames as labelled (red = 0 ns; white = 250 ns, blue = 500 ns). Cholesterol is shown relative to the SCTR (final frame at 500 ns).

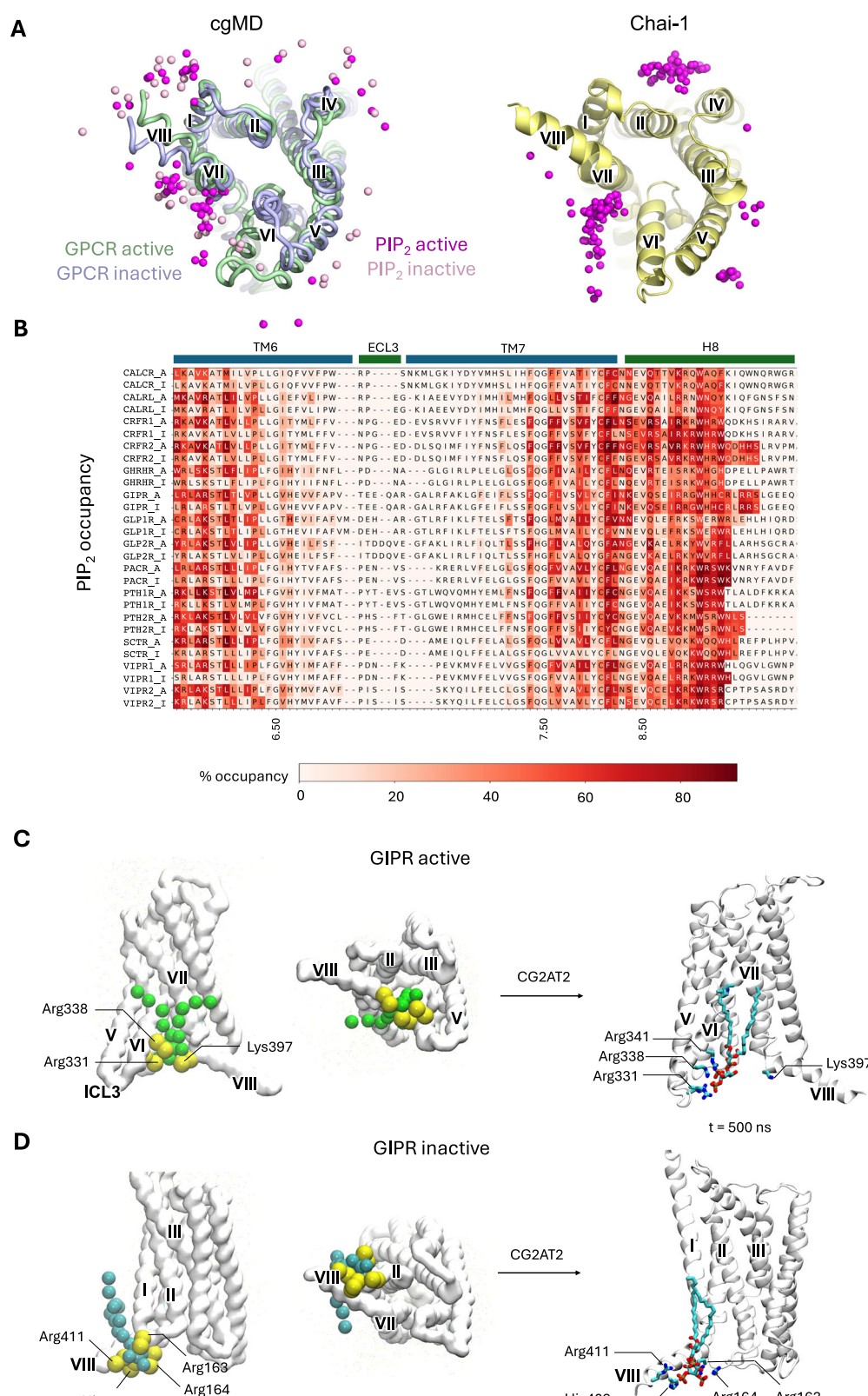

**Fig. 4 | Interactions of PIP$_2$ with class B1 GPCRs. A** Alignment of top ranked PyLipID PIP$_2$ interaction sites across all 15 receptors from cgMD simulation and Chai-1. The view shown is from the cytoplasmic side onto the base of the active (green) and inactive (blue) states of the CTR. Spheres represent PIP$_2$ headgroups for the top ranked interaction sites extracted from simulations of active (dark magenta) and inactive (light pink) states of all 15 class B1 GPCRs. Chai-1 predicted an active state for all 15 receptors. For each receptor, five Chai-1 models were generated, each containing three PIP$_2$ poses. This yielded 15 poses per receptor, and 225 poses overall. Only poses within the inner leaflet were considered. **B** PIP$_2$ occupancy mapped onto sequence alignment of class B1 GPCRs for regions TMH6-8. **C** Highest residence time PIP$_2$ site in the GIPR active state corresponding to TM6/7, and H8. Residues in contact with the PIP$_2$ are labelled. The final position of the PIP$_2$ lipid following 500 ns atomistic simulation is shown. **D** Highest residence time PIP$_2$ binding site in the inactive GIPR receptor, with residues in contact with the lipid headgroup shown. The final position of the PIP$_2$ lipid following 500 ns atomistic simulation is shown.

**Fig. 5 | Modulation of GLP1R and GIPR ECDs by GM3 in cgMD simulations. A** Motion of the ECD during 10 μs simulation time for the active and inactive states for GLP1R. The starting structure is shown as solid grey, whilst the ECD motion relative to the TM bundle is shown coloured by timestep (0–10 μs, blue-green-yellow). Phospholipid head-group beads are shown as transparent grey spheres and GM3 GL beads are shown in magenta. **B** Final 10 μs conformations for each repeat. GM3 lipids within 6 Å of the ECD backbone are shown in magenta for the final frame of the first repeat. **C** Histogram of the angle between the ECD and the TM bundle for the simulations with GM3 present. **D** ECD motion of GIPR during simulation, coloured as in (**A**). **E** GIPR final conformations for each run, with all GM3 lipids contacting the ECD in the final frames shown, coloured as in (**B**). **F** Histogram of ECD angles for GIPR, as described in (**C**).

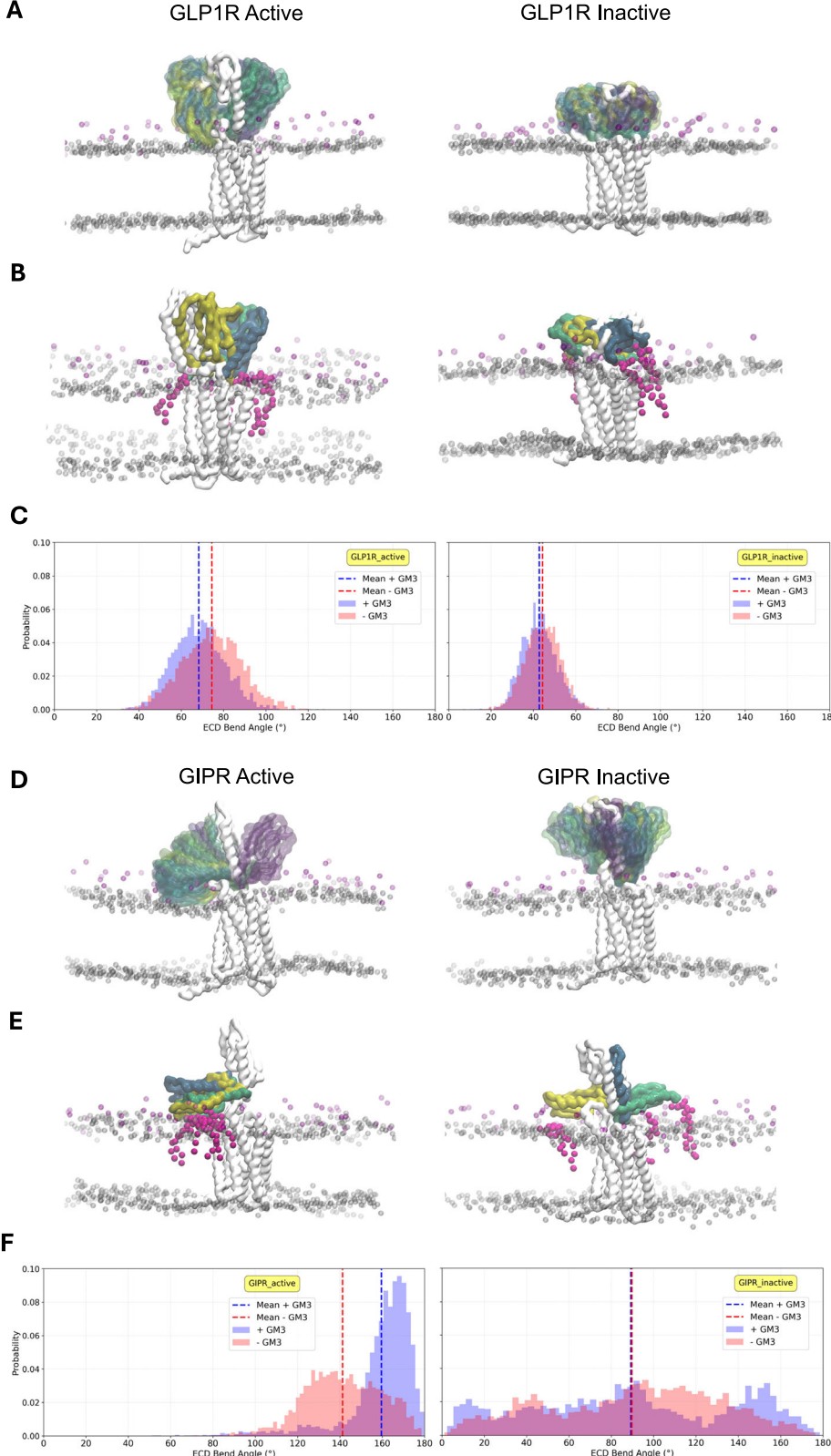

## The glycosphingolipid GM3 modulates the mobility of receptor ECDs

To characterise the interactions of ECDs with the plasma membrane, we analysed the ECD mobility of each GPCR in each state relative to the TM bundle (SI Fig. 16). These visualisations demonstrate that all B1 receptors in inactive and active states exhibit some degree of mobility of the ECD, but some receptors sample a much larger conformational space: GIPR, CRF1R and GHRHR (active states), and PAC1R, PTH1R and GLP2R (both states) showed the greatest mobility in our cgMD simulations.

To define the role of GM3 in ECD mobility, we chose to focus on two receptors in detail: GIPR, which showed the widest conformational variability in these simulations, and GLP1R which is representative of the overall B1 receptors, showing moderate mobility during our simulations (Fig. 5A, E). For the GLP1R and GIPR, we observed different modes of GM3 binding.

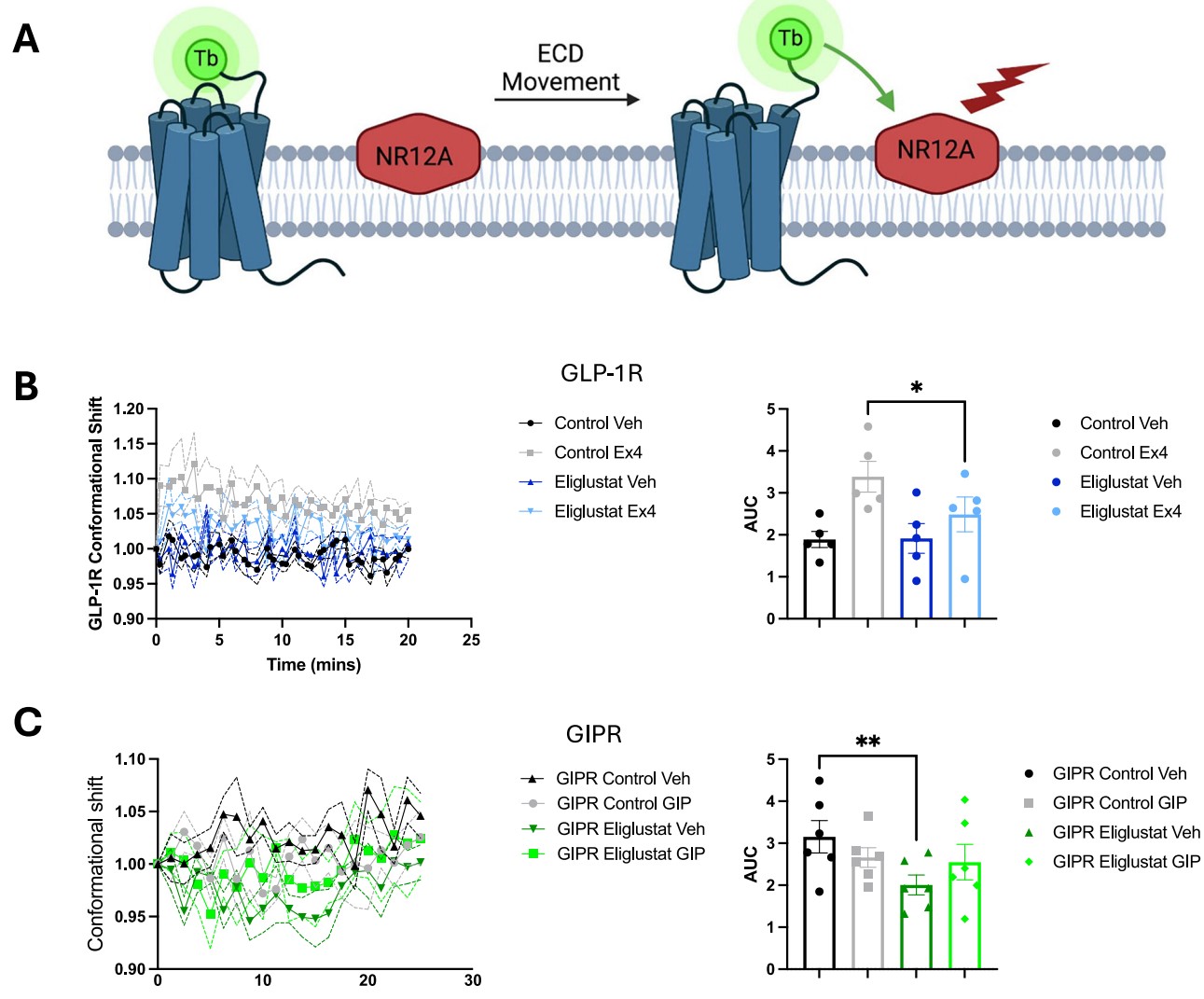

**Fig. 6 | TR-FRET ECD opening conformational shift assay. A** Schematic diagram showing the ECD—plasma membrane conformational shift assay: SNAP-tagged receptors are expressed in pancreatic β-cells to mimic native plasma membrane lipid composition, followed by Terbium (LumiTb, Tb) labelling of the SNAP-tag and NR12A labelling of the plasma membrane prior to TR-FRET analysis. Created with Biorender. **B, left** Conformational shift response (TR-FRET ratio) over time between SNAP-Lumi4Tb-labelled GLP1R and NR12A under Control or 300 μM Eliglustat conditions following stimulation with vehicle (Veh) or 100 nM exendin-4 (Ex4); (**B**, right) Area under the curve (AUC) for the TR-FRET kinetic response; $n = 5$ biological experiments, each with 4 technical repeats, $p^* = 0.0412$. **C**, left Conformational shift response (TR-FRET ratio) over time between SNAP-Lumi4Tb-labelled GIPR and NR12A under Control or 300 μM Eliglustat conditions following stimulation with vehicle (Veh) or 100 nM GIP; (**C**, right) Area under the curve (AUC) for the TR-FRET kinetic response; $n = 5$ biological experiments, each with 4 technical repeats, $p^{**} = 0.0091$. Error bars represent ± standard error of the mean.

For the active state GLP1R, GM3 bound tightly (maximum residence time 1525 ns) to the base of the ECD and the orthosteric ligand pocket, restricting the mobility of this domain (Fig. 5B). For the inactive state GLP1R, GM3 bound to the base of the ECD but did not interact with the orthosteric pocket, with a maximum residence time of 398 ns. For GIPR, GM3 was found to bind to the distal end of the ECD, forming contacts from the bulk membrane, with maximum residence times of 590 ns for the active state, and 718 ns for the inactive state (Fig. 5F).

To further assess the role of GM3 in ECD modulation, we ran comparative cgMD simulations with GM3 absent from the membrane. To quantify the ECD motion, we calculated the ECD bend angle[12] relative to the TM bundle (SI Fig. 17) for each state, with and without GM3 present in the membrane (Fig. 5C, G). For active state GLP1R, the absence of GM3 caused a small shift in the position of the ECD relative to the TM domain. For active state GIPR, a pronounced difference was observed, with the presence of GM3 restricting the motion of the ECD such that it remained membrane

interacting (Fig. 5F, G). The inactive state remained highly flexible with and without GM3, able to contact the membrane by forming two different states (Fig. 5F).

To evaluate the impact of using a coarse grained model (including an elastic network) on the observed ECD mobility, we converted the final frames from each cgMD replicate as the starting point for 500 ns atomistic simulations, with and without GM3 present in the membrane (SI Table 2). A comparison of ECD conformations sampled showed that these were largely determined by their initial conformations, on the 500 ns timeframe (SI Fig. 17). Accordingly, comparison of GM3 contacts showed similar trends for atMD and cgMD (SI Fig. 18).

To experimentally validate the observed role of GM3 in modulation of the ECD of GLP1R and GIPR in our simulations, we used a time-resolved Förster resonance energy transfer (TR-FRET) assay to measure receptor ECD proximity to the plasma membrane under vehicle and stimulated conditions (Fig. 6A). Experiments were conducted in the presence and

absence of the glucocerebroside synthase inhibitor Eliglustat to inhibit GM3 synthesis[50]. For GLP1R, stimulation with the agonist exendin-4 elicited a conformational rearrangement resulting in increased TR-FRET between the receptor ECD and the plasma membrane, an effect that we interpret as a proxy for ECD opening upon agonist binding. Additionally, a significant reduction in ECD opening was observed in the presence of Eliglustat, indicating that the ECD of GLP1R is more flexible when there is GM3 in the plasma membrane (Fig. 6B). Conversely, for GIPR, we observed that the ECD was less dynamic in the active (GIP-stimulated) compared to the inactive state (Fig. 6C), potentially due to the membrane-bound ECD open-state mode observed during the cgMD simulations. No significant difference in ECD opening between the untreated and the Eliglustat-treated conditions was observed for GIPR. Moreover, a higher basal state fluorescence indicated that the GIPR ECD is interacting with the membrane in the basal state more than GLP1R, consistent with our simulations (Fig. 5F). Our cgMD simulations partially agree with the TR-FRET results but we also find some discrepancies. For GLP-1R, simulations showed limited ECD dynamics in the inactive state under both 0% and 10% GM3 conditions (SI Fig. 15), which agrees with the experimental finding that there is no significant difference between the control and Eliglustat conditions. In the active state, while experiments suggest a significant reduction in ECD dynamics between the control and Eliglustat conditions, our simulations did not show a strong GM3-dependent effect on ECD dynamics. For GIPR, our cgMD simulations indicated that the ECD is most dynamic in the inactive state, showing a wide range of bend angles under both 0% and 10% GM3 conditions (SI Fig. 15). This supports the experimental interpretation that the GIPR ECD is contacting the membrane in the absence of ligand. However, we did not observe a significant reduction in ECD dynamics upon GM3 depletion in our simulations, as was seen experimentally. Taking the simulation and experimental results together, we observe state-dependent modulation of GIPR and GLP1R by GM3, and our results suggest that a fully closed inactive state, observed in the cryo-EM structure of GLP1R, may not exist for GIPR.

### Rare deep membrane interactions of POPC

Our global analysis highlighted high-residence time binding of POPC to PAC1R in the active state (SI Fig. 1). The same pattern of binding is also apparent for GHRHR in the active state, albeit at lower residence times (SI Fig. 1). We analysed both binding modes, which correspond to the same site at TMH3-4 (SI Fig. 19A, B), a deep membrane site recently identified as occurring frequently in a large scale GPCR simulation study of POPC membranes[44]. We found that in each case, in a single replicate, a POPC molecule entered this pocket from the intracellular leaflet, adopting a deep membrane binding mode with the POPC headgroup buried between TMH3-4 (SI Fig. 19A, B). In both pockets the POPC headgroup interacts with a buried Trp residue as well as interacting with the surface W4.50. A sequence comparison (SI Fig. 19C) indicates that this binding site may be present for all Class B1 receptors.

### Discussion

This study of class B1 GPCRs has identified trends in specific lipid interactions across the subfamily, highlighting avenues for further in-depth exploration. Key findings include the "deep membrane" binding mode of cholesterol which has been observed in multiple simulation studies at both coarse grained and atomistic resolution, but to the best of our knowledge has not been captured by experimental methods. We note that a recent X-ray crystallography and simulation study of aquaporin captured cholesterol in a deeper position between aquaporin tetramers than bulk membrane, but the polar headgroup does not access the centre of the core[51]. Our previous work on GLP1R has provided evidence that mutating a cholesterol binding site impacts downstream signalling and functional responses[18]. The results presented here indicate that the secretin receptor may be more tightly modulated by cholesterol, with distinct high residence time cholesterol binding sites in the active and inactive states potentially indicating a route to

modulating biased signalling of this receptor. The His211 identified in the long residence time active state site and the Ser231 of the inactive state site are shown in the gnomAD database[52] as variants of potential interest, with unknown pathogenicity.

Aside from the cholesterol deep binding mode, we also observe rare instances of POPC lipids inserting between TMH3-4 for PAC1R and GHRHR. POPC deep membrane binding has been recently described in a large scale GPCR simulation study using POPC-only membranes at atomistic resolution[44]. Our simulations provide some evidence that such sites may still be accessed by POPC in the presence of more complex membrane compositions. The GHRHR site was identified by visual inspection and comparison to the POPC occupancy plot for PAC1R, which indicates there is an opportunity for machine learning pattern recognition tools to be used to extract lipid binding sites from our datasets.

We found that, in a model plasma membrane, class B1 ECDs are highly mobile. Our analysis of GM3 ECD modulation of GLP1R and GIPR by both simulation and experiment demonstrate that GM3 has receptor-specific effects on ECD modulation, adding to the growing body of evidence that GM3 influences GPCR behaviour. Previous studies have shown that both GCGR and GHRHR are modulated by GM3[12,53]. Taking our simulation and experimental results together, our findings suggest that GM3 can modulate ECD–membrane interactions in a receptor- and state-dependent manner for GLP-1R and GIPR. Our simulation data indicates that this is a general feature of the class B1 GPCRs and may be a valuable pathway to explore for future therapeutic design efforts.

We additionally identify a highly conserved $PIP_2$ interaction site formed by TM6/7 and H8 *via* physics-based simulation. $PIP_2$ binding at this site shows a strong preference for the active over the inactive state, with the outward movement of TM6 exposing additional surface area and residues, as well as a 'groove-like' architecture into which a $PIP_2$ lipid may 'slot'. This site has previously been reported for isolated studies of the glucagon receptor (GCGR)[12,54]. The markedly high degree of conservation that we observe across all B1 receptors, the strength of interaction, and the location of this site adjacent to TM6, which undergoes the key movement seen during the inactive/active conformational change, is particularly intriguing. This raises the prospect that $PIP_2$ may modulate the kinetics of the transition and/or stabilise the active state, as has been seen for select class A GPCRs[53,55]. Similarly, binding of $PIP_2$ at this site in class B1 GPCRs may have a role in the recruitment of soluble intracellular signalling partners. In class A GPCRs, $PI(4,5)P_2$ binding at TM4 and TM1 has been shown to stabilise the active state, and enhance coupling to mini-$G\alpha_s$ over mini-$G\alpha_i$ and mini-$G\alpha_{12}$ by acting as a 'molecular glue' to bridge complex formation[17]. Functional experiments have also reported a role of $PI(4,5)P_2$ in enhancing β-arrestin recruitment for select GPCRs[11,42,56,57]. The $PIP_2$ site identified in the present study of class B1 GPCRs is particularly conserved across the subfamily compared to $PIP_2$ sites in class A GPCRs[17,42], and merits further investigation as to its possible functional consequences. We note that two antagonists of the GCGR have been resolved bound at this same interaction site (PDB ids: 5EE7 and 5XEZ)[58,59], raising the prospect of direct phosphoinositide / drug competition and synergistic binding effects. Similarly, a positive allosteric modulator (PAM) used to stabilise structures of GIPR in complex with the antidiabetic therapeutic tirzepatide (Mounjaro) (PDB id: 7RBT) binds proximal to the TM6/7-H8 site and may interact with lipids in this site[60].

Taken together, our comparison of cgMD simulations to the artificial intelligence-based predictions shows that interactions requiring model membrane environments or subtle conformational changes are not yet captured by structure prediction-based approaches, highlighting that the molecular simulation approaches remain a powerful tool in studies of membrane protein-lipid interactions. Tools such as aiida-gromacs, alongside efforts to standardise data in the simulation field, including MDVerse [61] and the Molecular Dynamics Data Bank (https://mddbr.eu), may enable training of AI models and improve predictive capacity within complex bilayer environments.

## Limitations

It is important to discuss the limitations of the present study. Our study was enabled by the availability of all 15 receptor cryo-EM structures, but these were mostly in the monomeric active state, whilst most inactive states had no experimental structures available. There is therefore a need for further class B1 GPCR structural studies to resolve inactive states, as well as intermediate states to enable the role of specific lipids throughout the full conformational cycle to be captured[62]. We used the Martini 3 forcefield which has been extensively employed for specific protein-lipid interactions. This has the advantage of stabilising inactive states in substrate and G protein-free conformation, but with the consequence that it limits the conformational space explored. This is particularly relevant in the observation of ECD motion, where the elastic network used to maintain protein structure will impact conformational mobility. Our simulations captured some elements of our experimental findings, but we may attribute differences to a combination or subset of: the different temporal and spatial scales; a single receptor in a membrane patch in the simulations compared to the full biological complexity in experiment; restricted sampling (including protein conformations) in Martini 3 simulations.

Sampling remains a challenge for coarse grained and atomistic simulation studies[63]. Our analyses of 10 μs and 100 μs timeframes showed comparable global properties across lipids, yet we observe instances of long residence time binding of cholesterol (to SCTR), and deep membrane POPC binding (PAC1R, GHRHR) events in individual replicates, limiting our ability to compute reliable error estimates[64]. This is particularly the case for PyLipID defined lipid site residence times, where different residues may be assigned to a given site across replicates. Our observations are intended to inform future experiments (and simulation studies), such as mutagenesis of the observed "deep membrane" cholesterol and POPC sites identified in SCTR, PAC1R and GHRHR.

There are multiple routes to extend this work, including experiments to validate the putative novel lipid binding sites described here, as well as confirming the family-wide modulation of the ECDs by GM3. We have made our set up and simulation files available to the community to build on this study. Future work will lead to building a more complete picture of class B1 GPCR signalling, by including heterodimerisation states[65,66], substrates, G protein/ β-arrestin complexes and post translational modifications[67].

## Materials and methods

### GPCR model building

B1 GPCRs structures in active and inactive states were taken from cryo-electron microscopy (cryo-EM) structures and AlphaFold multistate models deposited in GPCRdb[68]. For the inactive state, the GPCRdb models were used when there were no experimental structures available (13/15 inactive states). For active state structures, GPCRdb models were used for consistency when the Cα RMSD with the experimental structure was <0.8 Å. Above this cut off the ECD in GPCRdb models did not align with the experimental structure, and so the ECD from the experimental structure was used instead (SI Table 1). The low-confidence region in the N-terminal ECD and C-terminal tail were removed to match the existing experimental structures for all GPCRdb models used. Modeller v10.1[69] was used to add missing loops and the CHARMM-GUI PDB reader[70] was used to model any missing atoms in experimental structures. All models used in this study used the native human sequence. Further information about each model, along with the associated PDB accession codes[71] can be found in SI Table 1. [747576]

### Coarse-grained MD simulations

GPCR structures were converted to a coarse-grained representation using Martinize2[72] and the Martini 3 force field[22]. ElNeDyn elastic network restraints were applied to each GPCR using an elastic bond force constant of 500 kJ/mol/nm² and an upper cut-off of 0.9 nm[73]. The transmembrane region of GPCR was predicted using PPM 3.0 Web Server[74] and embedded into an asymmetric lipid membrane bilayer in a 25 x 25 x 19 nm³ box using the insane.py script[75]. The mammalian plasma membrane composition previously described[76] was used as follows: POPC (25%), DOPC (25%),

POPE (8%), DOPE (7%), GM3 (10%) and cholesterol (25%) in the upper leaflet, and POPC (5%), DOPC (5%), POPE (20%), DOPE (20%), POPS (8%), DOPS (7%), POPI(3,4)P₂ (10%) and cholesterol (25%). For the simulations of GLP1R and GIPR in a 0% GM3 membrane, POPC and DOPC were increased to 30% each in the upper leaflet with all other components the same. The most recent Martini 3 parameters for PI(3,4)P₂[77] and cholesterol[78] were used. The parameters for PI(3,4)P₂ were used as they correspond to the PIP₂ version used in previous Martini 2 studies[12,42]. We also ran comparative simulations using PI(4,5)P₂ for GLP1R and GIPR in both inactive and active states as test cases, with limited variation observed across these simulations (Fig. S20). Thus, at coarse grained resolution, we are not likely to resolve differences in binding of PI(3,4)P₂ and PI(4,5)P₂. Each system was solvated using Martini water[22] and 0.15 M NaCl to neutralise the system, followed by minimisation and equilibration steps using protocol files from CHARMM-GUI Martini Maker[79]. Briefly, the system was subjected to steepest descents energy minimisation, followed by six sequential NPT equilibration steps with restraints with decreasing force constant in each subsequent step from 1000 kJ/(mol nm²) to 50 kJ/(mol nm²) for the protein backbone and 200 kJ/(mol nm²) to 10 kJ/(mol nm²) for lipid headgroups.

Production run simulations were performed for 10 μs of cgMD. Three repeats were run for each active and inactive state, initiated from the same initial particle configurations, and random initial velocity seeds. A summary of simulations performed can be found in SI Table 1. The v-rescale thermostat (tau 1.0 ps)[80] and the Parrinello–Rahman barostat (tau 12.0 ps)[81] were used to maintain temperature (303.15 K) and pressure (1 bar) on all production runs. Nonbonded interactions were treated with reaction-field electrostatics with a cutoff of 1.1 nm[82]. All simulations were performed using GROMACS 2022.4 (https://doi.org/10.5281/zenodo.7323393). Lipid interaction profiles were calculated using PyLipID[30] with a cut-off of 0.7 nm to calculate lipid residence time and occupancy profiles across all three repeats. VMD[83] and PyMOL[84] was used for trajectory visualisation and analysis. 3D volumetric surfaces were generated with the VMD VolMap plugin[85]. For the longer timeframe simulations of SCTR, the same protocol was used except that the insane.py script was used to generate three independent starting configurations, and production runs were performed for 100 μs each.

### Atomistic simulations

The selected frames (shown in Table S2) from the coarse-grained simulations were backmapped to atomistic resolution using the CG2AT2 package[47], in which the protein and the membrane were backmapped. For GIPR, the last frame of each of the three simulation repeats was selected for backmapping, as these represent the two extremes of ECD motion (ECD bend angles ranging from 30° to 180°), as shown in Fig. S10. The same final frames were used for GLP-1R to ensure consistency. For SCTR, the selected frames corresponded to the bound pose with the highest residence time, identified using the PyLipID package[30].

The system was solvated using the CHARMM TIP3P water model[82] and 0.15 M of NaCl was added to give a neutral system. The system was then subjected to steepest decent energy minimisation steps, followed by six sequential NPT equilibration steps with decreasing restraints on the lipid headgroups and protein backbone.

The CHARMM36m forcefield was used[86]. All the simulations were performed using GROMACS 2022.4 (https://doi.org/10.5281/zenodo.7323393). 500 ns of production simulations were performed for each system, and the temperature was maintained at 303.15 K using the Nose-Hoover thermostat[87,88] and the pressure at 1.0 bar using the semiisotropic Parrinello-Rahman barostat[81] (τT = 1.0 ps and compressibility = 4.5 x 10⁻⁵ bar⁻¹). The periodic boundary condition was applied and a time step of 2 fs was used. The Particle Mesh Ewald (PME) method[89] was used for the long-range electrostatic treatment. The LINCS algorithm was used to constrain H-bonds[90].

### Reproducible simulation protocols with aiida-gromacs

These simulations were produced using an automated workflow with each step for setting up the system captured using aiida-gromacs,

allowing others to reuse or extend our dataset to more GPCRs. An exemplar trajectory and starting files are available on zenodo (https://doi.org/10.5281/zenodo.14359056)[91] along with the single AiiDA archive file, which contains all the steps involved to reproduce the simulation methodology described above (https://doi.org/10.5281/zenodo.14728414)[92] Along with the raw files and simulation metadata, a Jupyter Notebook tutorial (https://github.com/PSDI-UK/aiida-gromacs/tree/master/notebooks, https://aiida-gromacs.readthedocs.io/en/latest/tutorials/PTH2R_CGMD.html) is available, showcasing the simplicity in provenance capture using aiida-gromacs in this study.

### AI models
The AI models generated for this study utilized the Chai-1 multi-modal foundation model described in ref. 27(https://github.com/chaidiscovery/chai-lab/). Chai-1 employs a similar model architecture and training strategy to that of AlphaFold 3[25], and is reported to perform well across a variety of protein-ligand prediction tasks[27]. In the present work, protein-lipid models were generated using the standardized web interface (https://lab.chaidiscovery.com/). No custom restraints were employed. The multiple sequence alignment (MSA) option was set to active for enhanced accuracy. The model we utilized had a training cutoff date of 2021-01-12 and will not have seen experimental GPCR-cholesterol structures after this date. Primary sequences of the same class B1 constructs used in MD simulations were extracted and used as input for the Chai-1 model, along with the SMILES strings for cholesterol, and PI(4,5)$P_2$. SMILES strings were obtained from PubChem[93]. The PI(4,5)$P_2$ variant used was the 1-stearoyl-2-arachidonoyl, which is the most common tail saturation pattern for PI(4,5)$P_2$ in human cells[94]. Five AI models were generated for each cholesterol run, with three cholesterol molecules in each run. This yielded a total of 15 cholesterol poses for each receptor, and 225 cholesterol poses globally across all receptors. PI(4,5)$P_2$ predictions followed the same protocol.

### Conformational assay by TR-FRET
INS-1 832/3 pancreatic β-cells stably expressing SNAP/FLAG-hGLP1R or SNAP/FLAG-hGIPR were pre-incubated in suspension with 40 nM of SNAP-Lumi4-Tb (SSNPTBC, Cisbio), a lanthanide fluorophore probe for TR-FRET, or 40 nM SNAP-Lumi4Tb plus 300 μM Eliglustat, an inhibitor of GM3 gangliosides[95] for 1 h at 37 °C in full culture media. Cells were pelleted and washed with HBSS and then resuspended in 100 nM NR12A, a Nile red-based plasma membrane fluorescent probe[96], diluted in HBSS, incubated for 5 min, and then seeded unto white opaque 96-well half-area plates. TR-FRET was carried out as previously described[97]. Briefly, a baseline reading was taken for 5 min using a Flexstation 3 plate reader with an excitation of 335 nm, emission of 490 nm and 590 nm at 37 °C. The cells were stimulated with vehicle or either 100 nM Exendin-4 (GLP1R) or 100 nM GIP (GIPR) and the plate was read for a further 20 min at 37 °C. Receptor conformational shift was calculated as the ratio of the fluorescent signal of both wavelengths (590/490 nm) after normalising to basal signal. Area under the curve (AUC) for the TR-FRET kinetic response was calculated from $n = 5$ individual experiments with 4 technical repeats per experiment. Data presented as mean ± Standard Error of the Mean (SEM) and analysed using one way ANOVA with Šídák's multiple comparisons test $p* < 0.05$. Data was collected using SoftMax Pro software on a Flexstation Plate reader (Molecular Devices) and copied into Excel (Microsoft). Data was analysed using Prism 10 (GraphPad).

### Statistics and reproducibility
Each molecular dynamics simulation condition had at least three independent replicates. Per residue lipid occupancy mean values and standard deviations were calculated from three replicates. The simulations are reproducible using the same settings and input files provided. TR-FRET assays used 5 individual experiments with 4 technical replicates per experiment.

### Reporting summary
Further information on research design is available in the Nature Portfolio Reporting Summary linked to this article.

### Data availability
All data are available in the main text or the supplementary materials. Additional data including raw data for plots presented has been deposited on Zenodo: https://doi.org/10.5281/zenodo.14359056[91]; https://doi.org/10.5281/zenodo.14728414[92].

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

## Acknowledgements

We thank Joseph Barritt for helpful discussion. This work was supported by the following grants: UKRI Future Leaders Fellowship (MR/Y01975X/1): S.L.R.; MRC Project Grant (MR/X021467/1): A.T. (Y.M.); Wellcome Trust Discovery Award (301619/Z/23/Z): A.T. (A.I.O.), S.L.R. (G.H.)

## Author contributions

K.W.C., G.H., S.L.R., A.T., J.G.R., J.K.. Methodology: K.W.C., L.W., J.K., A.T.. Investigation: K.W.C., L.W., G.H., S.L.R., Y.M., A.I.O.. Visualization: K.W.C., L.W., G.H., S.L.R., A.T., A.I.O., Y.M.. Supervision: G.H., A.T., J.G.R., S.L.R.. Writing—original draft: K.W.C., G.H., J.K., J.G.R., A.T., S.L.R.. Writing —review & editing: L.W., A.I.O., J.K., J.G.R., G.H., A.T., S.L.R.

## Competing interests

The authors declare no competing interests.
