## [Transparent Peer Review file · Communications Biology]

Human class B1 GPCR modulation by plasma membrane lipids.

Corresponding Author: Dr Sarah Rouse

Version 0:

Reviewer comments:

Reviewer #1

(Remarks to the Author)

Chao and coworkers studied the localization of lipids around B1 GPCRs, by using coarse-grained molecular dynamics simulations. They considered multiple members of this protein family in both their active and inactive states and carried out multi-microsecond long MD simulations, when these proteins were embedded in multi-component lipid bilayers resembling the lipid composition of the plasma membrane. In addition, they predicted the position of lipids around these proteins by an AI foundation model. They also examined the impact a specific lipid (GM3) has on the mobility of the extracellular domain of these proteins, data that was accompanied by FRET experiments.

I find this is a very interesting study which sheds lights into possible general lipid-protein interaction patterns within the B1 GPCR family. I appreciate the effort to make the data available, reusable and extendable. I have several comments about the methodology, the contextualization of the results in the light of existing data. Furthermore, the way the data was presented was not always clear.

The amount of sampling is impressive (of ~30 us per protein for 15 proteins in two states). However, as the authors point out in the limitations section, Martini3 may limit the conformational space explored. Therefore, it remains unclear whether 30 us/protein was sufficient to fully capture localization of lipids around these proteins. Uncertainties of the shown occupancies, lipid contacts and lipid residence times, and other quantities presented in the text, should have been included to clarify that.

On the mobility of the extracellular domain (figure 5 and text referring to it), It is unclear whether this is modulated by GM3, given that the protein conformation has been restricted at either the active/inactive state by the elastic network model. All-atom MD simulations, of the two specific selected systems, i.e. GLP1R and GIPR, could have been used to confirm the modulation hypothesis.

Localization of lipids around GPCRs, in particular that of cholesterol, has been extensively investigated by MD simulations. In particular, in a recent study (Ref. 43 PMID: PMC7175695) a similar coarse-grained simulation campaign has been carried out covering different types of GPCRs. How much do the findings reported here overlap and/or expand these previous findings?

The observation of deep positions for cholesterol is very interesting. This has been previously observed both by simulations and experiments for another membrane protein, i.e. an aquaporin (PMCID: PMC11368405). In this case, cholesterol had to sandwich in between two aquaporin tetramers in order to get in such deep position. For the studied GPCRs here this does not seem to be necessary. The authors should discuss their findings in the light of that previous data, even though it goes beyond GPCRs.

Also, it remains unclear whether the deep positions were rather transient or long-lived. This is very important as cholesterol has been observed to flip flops around membrane proteins in the micro-second scale suggesting deep positions may be transient (see e.g. <https://doi.org/10.1021/acs.jctc.8b00933> , <https://doi.org/10.1021/jp108166k> , <https://doi.org/10.1371/journal.pcbi.1005240>).

- The way some of presenting the data was not optimal. Here some general comments:

- The deep cholesterol data shown in figure 3C could be further visualized in a more quantitative fashion by showing histograms of the position normal to the membrane of a reference group, e.g. the hydroxyl group.
- Beyond the representation of static structures, time-averaged 3-dimensional density maps could help the visualization of lipid positions, as done, e.g. here in this study (PMID: 32954729) with GROmaps.
- Some plots have extremely small (unreadable) fonts.
- Some plots lack units. For instance what are the units of the lipid residence times?
- the different scales in the color maps make difficult to compare them (see for instance Fig 2).
- L 150 -159: it is very hard to see the differences between active/inactive of the TM6 region compared to other regions in Fig 2. I recommend to post process that data further to more clearly show the difference in these properties as function of the protein conformation (active or inactive).
- The authors emphasize on the use of aiida-gromacs to capture full provenance for the set-up of simulations in complex plasma membranes. This is highly appreciated. I would like to see more on the advantages of this setup compared to a traditional simulation setup and some indications how the data presented here can be accessed.
- In the methods it was not mentioned whether neutralizing ions were also added, apart from the 0.15 NACL concentration.

Reviewer #2

(Remarks to the Author)

In their manuscript "Human class B1 GPCR modulation by plasma membrane lipids", Kin Chao and colleagues study the interactions of cholesterol, PIP2, and GM3 on 15 human class B1 GPCRs mainly using coarse-grained simulations. The manuscript is well written and easy to follow. However, while the simulations and analyses are mostly performed to present-day standards, I would expect a more comprehensive characterisation of the observed phenomena, especially considering how fast coarse-grained simulations can be performed nowadays.

I provide suggestions regarding this below, in no particular order.

-While I appreciate the use of multiple replicate simulations (a practise still often omitted), I wonder if the 10 μ s simulations are adequate for lipid mixing in the large membrane setup with lots of cholesterol. Could the authors estimate, e.g., the average distance of lipids travelled based on their diffusion coefficients.

-Related to the previous point, discarding only 200 ns of the simulation time seems too little. The authors find that some lipids bind for microseconds, which indicates that the three simulations initiated from then same structure are likely not uncorrelated after the equilibration time. Solutions here would be extending the simulations, extending the equilibration period, or running one replica starting from a completely independent starting structure. With the automated pipelines, this should not be a major task.

-The coarse-graining naturally limits Martini's ability to describe conformational entropy (which is an important parameter in lipid-protein interactions, notably with polyunsaturated fatty acids), which is then incorporated into the LJ interactions. Martini also has a limited ability to describe electrostatic interactions. These factors could significantly affect the binding of PIP2 with polyunsaturated fatty acid chain and a highly-charged headgroup. To this end, I would expect the authors to validate (at least some) of the results involving PIP2 with atomistic models by fine-graining suitable PIP2-bound structures observed in coarse-grained simulations. Optimally, these validation simulations could involve biased simulations to extract PIP2 binding affinities.

-The authors used Chai-1, as it provides "AF3-like accuracy". I wonder why AF3 was not used for comparison?

-The font sizes in Figures are often way too small. For example, it's impossible to read the labels in Fig. 2.

-From the general patterns of lipid interactions, can we learn something about interaction motifs? For example, if some members of the B1 GPCRs do interact and others don't, does this correlate somehow with their sequences?

-The difference of "occupancy" and "contacts" in Fig. 2 is not clear from the results section.

-In Fig. 2 (or in the SI), could the authors show a negative control, i.e. data for a lipid that doesn't have specific interactions.

-It would also be revealing to see the overall tendencies of the proteins to interact with different lipid classes, visualised by, e.g., a bar graph of normalized contacts of all the lipid classes in the simulation system.

-Is the cholesterol binding site near TM6 a non-annular one, so that cholesterol only fits there in the active conformations? Or is there another justification for the interaction observed mainly in the active conformation?

-What are "summed [...] residence times"? I think it would be more useful to provide the typical interaction times as these can be compared to other studies.

-For cholesterol-W^A(4.50) interaction, is there a specific pair of Martini beads in cholesterol and tryptophan that has a

particularly strong LJ interaction? Or do tryptophan and cholesterol adapt a specific interaction mode? Or do the neighbouring amino acids (and the protein shape) play a role here?

-It could be good to point out that cholesterol accesses the membrane core a lot in Martini simulations containing lipids with (poly-)unsaturated acyl chains of lipids. This is due to the repulsion of cholesterol by the double-bond beads that is required to capture liquid-liquid phase separation of ternary lipid mixtures. In atomistic simulations, this effect is significantly smaller.

-Fig. 3 again has too small fonts, panel B is especially hard to read. Panel C, on the other hand, is hard to grasp. I suggest to use either more contrasting colours than yellow & orange, or better yet, extract volumetric density maps of the occupancies.

-How well does PIP2 binding (Fig. 4A) correlate with protein surface charge?

-The text in Fig. 4B is unreadable.

-How much does the elastic network affect the ECD dynamics? Does the degree of mobility correlate with the amount of elastic network springs connecting ECD to the TM bundle?

-With all replicas starting from the same structure, it is very hard to assess the significance of the effect of GM3 on ECD conformation. I think this requires either validation using all-atom simulations, with coarse-grained biased simulations, or unbiased coarse-grained simulations using fully independent replicas, preferably with varying amounts of GM3 present in the membrane.

-Does the TR-FRET agree with the simulations? If I got it right, in experiments with GLP1R, activation leads to more interaction of ECD and plasma membrane, whereas in simulations the tilt angle is larger for the active case (ECD stands more upright). Also, Eligustat has no effect on GIPR, yet in simulations GM3 has a larger effect for GIPR.

-If PPM 3.0 was used to resolve the membrane positioning of the GPCRs, why does the workflow in Fig. 1 refer to gmx confms, which seems obsolete in this case? Moreover, are the steps performed in CHARMM-GUI (which to my knowledge cannot be scripted) integrated in the aiida approach? Fig. 1 omits them.

-The description of simulation methodology is incomplete. For example, information on the handling of non-bonded interactions is missing.

Version 1:

Reviewer comments:

Reviewer #1

(Remarks to the Author)

The authors have adequately addressed my questions and concerns. The manuscript has greatly improved with the the new additional control simulations and analyses. The only aspect that is still missing relates to the lines 341-342 of page 17. They wrote "Accordingly, comparison of GM3 contacts showed similar trends for at MD and cgMD (Fig. S18 B,C)." However, Fig S18 does not contain panels B and C. Thus, it is not possible to discern whether the GMX3 contacts showed similar trends.

Reviewer #2

(Remarks to the Author)

The authors have addressed my comments to an adequate degree, and the manuscript is now of suitable quality for publication.

I still recommend the authors to increase font sizes in all figures. At least in their current (raster) format, the text is often unreadable.

Response to Reviewers' Comments: [COMMSBIO-25-0310-T]

General:

We thank the reviewers for their helpful comments and suggestions. Our detailed responses are given below and we have highlighted changes in the text of the manuscript.

We have also:

- Corrected throughout the manuscript the PIP2 parameters used –the conventionally used PIP2 (e.g. PMID 32272057) corresponds more closely to PI(3,4)P₂ than PI(4,5)P₂.
- Included references to relevant studies that have been released during this revision process, including a large scale atomistic MD simulation study of deep membrane POPC pockets across GPCRs (PMID: 40016203).

Reviewer #1 (Remarks to the Author):

Chao and coworkers studied the localization of lipids around B1 GPCRs, by using coarse-grained molecular dynamics simulations. They considered multiple members of this protein family in both their active and inactive states and carried out multi-microsecond long MD simulations, when these proteins were embedded in multi-component lipid bilayers resembling the lipid composition of the plasma membrane. In addition, they predicted the position of lipids around these proteins by an AI foundation model. They also examined the impact a specific lipid (GM3) has on the mobility of the extracellular domain of these proteins, data that was accompanied by FRET experiments.

I find this is a very interesting study which sheds lights into possible general lipid-protein interaction patterns within the B1 GPCR family. I appreciate the effort to make the data available, reusable and extendable.

- **Author reply:** We are pleased the reviewer finds the work to be very interesting, and appreciates the high degree of sampling and data availability standards.

I have several comments about the methodology, the contextualization of the results in the light of existing data. Furthermore, the way the data was presented was not always clear.

The amount of sampling is impressive (of ~30 μ s per protein for 15 proteins in two states). However, as the authors point out in the limitations section, Martini3 may limit the conformational space explored. Therefore, it remains unclear whether 30 μ s/protein was sufficient to fully capture localization of lipids around these proteins.

- **Author reply:** The author raises an important point. Convergence and sampling in simulations is a critical and often overlooked aspect as we have written about previously (PMID: 27807980). In the present study we chose to use Martini3 coarse-grained (CG) simulations to allow sufficient sampling of lipid on/off events, which are not readily accessible by atomistic simulations. To assess whether 10 μ s is sufficient to capture the key protein-lipid interactions, we have added three new independent 100 μ s cgMD simulations of the SCTR receptor, each initiated from distinct lipid arrangements. The new **Fig. S5** presents the lipid occupancy calculated for 10 μ s vs 100 μ s. We also compare volumetric densities using the VMD VolMap tool (new **Fig. S6**) for 10 μ s and 100 μ s, revealing similar occupancy patterns between the two. We also include analysis of each repeat independently to generate error estimates for 10 vs 100 μ s

(**Fig. S8**) The 100 μ s results are consistent with those from the 10 μ s simulations, indicating that 10 μ s is sufficient to capture key interactions on a per-residue basis. Given the nature of our study in assessing a global view of a large group of proteins, we believe that a 10 μ s simulation length offers a good balance between sampling and computational efficiency and is well above the level of sampling for state-of-the-art simulations of related lipid-protein interactions with e.g. GPCRs – simulation study of GCGR ECD motions (already cited in the ms PMID: PMC7376093), lipid scramblase simulations (PMCID: PMC11001264). We have summarised our comparative analyses of 10 μ s vs 100 μ s simulations (**Page 7, lines 146-154**), and added further discussion on sampling limitations for both cgMD and atomistic simulations (**Page 26, lines 502-510**).

- As the reviewer notes, while Martini3 enhances sampling of certain aspects such as lipid diffusion, it limits sampling of e.g. protein conformational space due to the necessary imposition of an elastic network. Thus such simulations model lipid-protein interactions with a given conformational state of each receptor. We take account of this by running simulations of both the active and inactive conformations of each receptor, which are thought to be the two principal biological relevant states. Certainly it is true that other conformational states, yet to be structurally determined, may occur in cells under specific conditions, with which other interesting lipid interactions may occur. We have added a nod to this on **Page 25-26, lines 489-491**.

Uncertainties of the shown occupancies, lipid contacts and lipid residence times, and other quantities presented in the text, should have been included to clarify that.

- **Author reply:** We have now computed the PyLipID occupancy analyses per residue/state/receptor/replicate for cholesterol/PIP2/GM3/POPC to generate error estimates. A total of 120 plots are generated (15 GPCRs \times 2 states \times 4 lipids). All plots and raw data are uploaded to the Zenodo repository (<https://zenodo.org/records/16788845>). We also include the plots for SCTR in the Supporting Information (**Fig. S8**) as part of our analyses of 10 μ s 100 μ s timeframes. Analysis of residence time uncertainties for PyLipID defined sites is more complicated as different residues can be included in a given site across repeats - we have included a discussion section on this aspect (including sampling of rare events) on **Page 26, lines 502-510**. We have also reworded our description of the long residence time SCTR sites to acknowledge this (**Page 12, lines 239-243**).

On the mobility of the extracellular domain (figure 5 and text referring to it), It is unclear whether this is modulated by GM3, given that the protein conformation has been restricted at either the active/inactive state by the elastic network model. All-atom MD simulations, of the two specific selected systems, i.e. GLP1R and GIPR, could have been used to confirm the modulation hypothesis.

- **Author reply:** We agree this is an important point. To evaluate the ECD mobility observed in cgMD further, we have added new 500 ns atomistic MD simulations for both GLP1R and GIPR in the presence and absence of GM3, with three independent repeats for each condition backmapped from the last frame of each cgMD replicate (**Page 17, lines 336-342**). The ECD motion from these simulations is plotted in the same format as in Fig. 5 of the main text and is shown in new **Fig. S18**. The results are broadly consistent with those in Fig. 5. However, due to the limited sampling inherent in atomistic simulations, the observed ECD motion is highly dependent on the starting structure. While it remains a significant challenge to fully characterise the effect of GM3 on ECD mobility due to sampling issue, our combined coarse-grained and atomistic simulations, along with experimental assays, provide preliminary evidence supporting a potential modulatory role of GM3 across the class B1 family, building on previous isolated studies of GCGR and GHRHR. Further work will be necessary to conclusively determine the nature and mechanism

of this modulation. We have expanded our discussion to explicitly note the possible effect of the elastic network on **Page 26, lines 495-496**.

Localization of lipids around GPCRs, in particular that of cholesterol, has been extensively investigated by MD simulations. In particular, in a recent study (Ref. 43 PMID: PMC7175695) a similar coarse-grained simulation campaign has been carried out covering different types of GPCRs. How much do the findings reported here overlap and/or expand these previous findings?

- **Author reply:** The recent study referenced is a valuable one. They also observed long-time frame deep membrane cholesterol binding between TMH4-5 for AT₂R. The paper concerned focuses primarily on global analyses of the lipid interactions - for the subset of class B1 structures simulated (GLP1R, GCGR, and CLR all in the inactive state). We cannot access the raw data, but we have compared the broad trends in residue-level occupancy heatmaps. PIP2 interactions are highest for these three receptors in the inactive state at TM6-8, consistent with our findings. There are several differences between our study and the referenced Tieleman group GPCR study – we have used Martini 3 whereas the previous study uses Martini 2; we simulate both active and inactive states; their study uses the TMH bundle, whilst ours includes the ECDs. The ProLint server has been offline during this revision process, so we are unable to generate identical analyses to compare these in more depth. We hope it will be restored so that any reader can make use of it to fully compare the trajectories deposited on Zenodo. To summarise, our study describes the state-dependent lipid interactions of the entire class B1 subfamily using the new state-of-the-art Martini 3, whilst allowing broad comparison with prior Martini 2 simulations of the inactive states of a subset of three of the receptors.

The observation of deep positions for cholesterol is very interesting. This has been previously observed both by simulations and experiments for another membrane protein, i.e. an aquaporin (PMCID: PMC11368405). In this case, cholesterol had to sandwich in between two aquaporin tetramers in order to get in such deep position. For the studied GPCRs here this does not seem to be necessary. The authors should discuss their findings in the light of that previous data, even though it goes beyond GPCRs.

- **Author reply:** We thank the reviewer for highlighting this work – we have added this reference to the Discussion (**Page 23, lines 421-424**). We note that the cholesterol in this case is “deeper” than the bulk cholesterol, but the orientation remains such that the hydroxy group is not in the membrane core.

Also, it remains unclear whether the deep positions were rather transient or long-lived. This is very important as cholesterol has been observed to flip flops around membrane proteins in the micro-second scale suggesting deep positions may be transient (see e.g. <https://doi.org/10.1021/acs.jctc.8b00933>, <https://doi.org/10.1021/jp108166k>, <https://doi.org/10.1371/journal.pcbi.1005240>).

- **Author reply:** We agree that an overview of the residence times is helpful - we used a recently published atomistic MD GPCR study of POPC deep membrane binding (<https://doi.org/10.1038/s41467-025-57034-y>) to define short-, medium- and long-lived interactions, and use this to compare the top deep membrane and canonical cholesterol binding modes. We observe that the distribution is shifted towards longer time frames for the deep membrane modes, whereas the canonical binding orientations include more high occupancy, transient interactions. We have included this in the revised manuscript (**Page 10, lines 200-207**).

Related to this point we have investigated the deep membrane cholesterol binding mode at atomistic resolution. To do so, we backmapped three representative frames with the cholesterol molecule with the highest residence time for SCTR (Fig. 3C,D) to an atomistic representation and performed 500 ns atomistic simulations.

- In the backmapped AT-MD simulations, this perpendicular-bound cholesterol dissociated during the minimisation and equilibration steps for 2 repeats, and within the first 15 ns (partial dissociation) -200 ns (complete dissociation) for the third replicate. This could be due to the CG-to-AT backmapping scheme, which can introduce steric clashes or suboptimal atomic arrangements. However, following the dissociation, a different cholesterol molecule spontaneously sampled this perpendicular binding site in the latter half of the AT-MD simulation involving the same residues identified in the cgMD simulation. This provides further support of the observed binding mode and has been incorporated into the manuscript (Fig 3F new panel) and Page 13, lines 252-267. **Additional panels in Fig 3:**

From Fig. 3 Cholesterol interactions with class B1 GPCRs F) Cholesterol spontaneously entering the binding pocket during atMD simulation, with positions at given simulation frames as labelled (red = 0 ns; white = 250 ns, blue = 500 ns). Cholesterol is shown relative to the SCTR (final frame at 500 ns).

- The way some of presenting the data was not optimal. Here some general comments:

- The deep cholesterol data shown in figure 3C could be further visualized in a more quantitative fashion by showing histograms of the position normal to the membrane of a reference group, e.g. the hydroxyl group.

- **Author reply:** We thank the reviewer for this suggestion. We have now computed the headgroup density (using the ROH bead for Martini2 and the O3 atom for atomistic charmm36m cholesterol) for SCTR, GIPR and GLP1R at cg and atomistic resolution, and presented the results in Fig. S14. The cgMD profiles across different GPCRs are broadly similar, with density for cholesterol ROH groups inside the membrane center region across all GPCRs and states (Page 13, lines 255-258). For the atMD simulations, the presence of cholesterol headgroups in the center of the membrane is lower, but non-zero, and we observe cholesterol accessing the membrane core in our SCTR atomistic simulations (new Fig 3F panel).

- Beyond the representation of static structures, time-averaged 3-dimensional density maps could help the visualization of lipid positions, as done, e.g. here in this study (PMID: 32954729) with GROMaps.

- **Author reply:** We thank the reviewer for the suggestion. We agree that 3D volumes or surfaces can be useful. Due to the large amount of data presented in this study, we have opted to show the aligned occupancy sequence heatmaps in Figures S2–S4, which we believe are

more informative for guiding future studies with residue-level information, including mutagenesis experiments to validate these sites. We have also presented the top lipid binding sites along with representative binding poses (3D structures) for each state in Fig. S7,S8 as well as sharing the PyLipID outputs on zenodo. We believe this approach allows interested readers to easily access and interpret the key lipid interaction patterns. We have included VolMap 3D surfaces for SCTR (**new Fig. S6**) to allow a visual comparison of the binding sites for the 10 vs 100 us, and for a newly identified POPC binding site (**new Fig. S19**).

- *Some plots have extremely small (unreadable) fonts.*

- **Author reply:** All main text figures have been adjusted to improve readability.

- *Some plots lack units. For instance what are the units of the lipid residence times?*

- **Author reply:** We apologise for these omissions. Relevant figures and text have been corrected (including units for Fig. 2).

- *the different scales in the color maps make difficult to compare them (see for instance Fig 2).*

- *L 150 -159: it is very hard to see the differences between active/inactive of the TM6 region compared to other regions in Fig 2. I recommend to post process that data further to more clearly show the difference in these properties as function of the protein conformation (active or inactive).*

- **Author reply:** We have used different scales as a necessity, as each lipid has differing levels of occupancy and residence time. Thus the patterns for weak interactors would be obscured were the scales to be made uniform. We have further clarified the data through simplification of Fig 2. and addition of mean properties (**new Fig. S1**), alongside the residue-level heatmaps in the SI.

- *The authors emphasize on the use of aiida-gromacs to capture full provenance for the set-up of simulations in complex plasma membranes. This is highly appreciated. I would like to see more on the advantages of this setup compared to a traditional simulation setup and some indications how the data presented here can be accessed.*

- **Author reply:** We have now added the zenodo links within our revised manuscript (**Page 30, lines 599, 601**). We are pleased to note that our two aiida datasets have been downloaded over 200 times.

- *In the methods it was not mentioned whether neutralizing ions were also added, apart from the 0.15 NaCl concentration.*

- **Author reply:** We thank the reviewer for pointing this out. The addition of neutralizing ions alongside the 0.15 M NaCl concentration has now been included in the Methods section. (**Page 28, line 553**).

Reviewer #2 (Remarks to the Author):

In their manuscript "Human class B1 GPCR modulation by plasma membrane lipids", Kin Chao and colleagues study the interactions of cholesterol, PIP2, and GM3 on 15 human class B1 GPCRs mainly using coarse-grained simulations. The manuscript is well written and easy to follow.

- **Author reply:** We are pleased the reviewer finds the manuscript to be well written and easy to follow.

However, while the simulations and analyses are mostly performed to present-day standards, I would expect a more comprehensive characterisation of the observed phenomena, especially considering how fast coarse-grained simulations can be performed nowadays.

I provide suggestions regarding this below, in no particular order.

-While I appreciate the use of multiple replicate simulations (a practise still often omitted), I wonder if the 10 μ s simulations are adequate for lipid mixing in the large membrane setup with lots of cholesterol. Could the authors estimate, e.g., the average distance of lipids travelled based on their diffusion coefficients.

- **Author reply:** We have addressed this point by adding longer timeframe cgMD control simulations of SCTR in both active and inactive states. These were built from different initial lipid arrangements (100 μ s, 3 repeats). For each independent repeat, we calculated the mean square displacement (MSD) of 5 lipid types (CHOL, DPG3, PIP2, POPC, and POPE) using *gmx msd*. In **new Fig. S7** we show the MSD of the lipids for the 10 μ s vs 100 μ s datasets (3 repeats each) with a linear line fitted for the 10-90% data range, which was used to obtain the diffusion coefficients (D) using the Einstein relation: $D = \text{MSD} / (4 \times t)$ for 2D diffusion. The D values were obtained by dividing the slope by 4. The results of D are shown in Fig. S7 for the 5 lipids comparing the 10 μ s vs 100 μ s datasets, with the standard deviation shown. The D values obtained were consistent with the Martini 3 lipid diffusion coefficients studied elsewhere (<https://chemrxiv.org/engage/chemrxiv/article-details/67652de681d2151a02594f8d>). From the second figure, we observe a high degree of similarity between 10 μ s and 100 μ s simulations, with percentage differences ranging from -7.65% to 17.29% and no statistically significant differences (all $p > 0.05$). These results demonstrate that 10 μ s simulations provide comparable diffusion characteristics to longer simulations. Crucially, increasing sampling (cf. increasing mean square displacement) by an order of magnitude did not lead to a difference in lipid-protein interaction patterns as measured by occupancy (Fig S6, 8). We further note that 3 x 10 μ s is in-keeping with the state-of-the-art seen in simulation works of lipid interactions with other membrane protein families (eg PMID 32610088).

-Related to the previous point, discarding only 200 ns of the simulation time seems too little. The authors find that some lipids bind for microseconds, which indicates that the three simulations initiated from then same structure are likely not uncorrelated after the equilibration time. Solutions here would be extending the simulations, extending the equilibration period, or running one replica starting from a completely independent starting structure. With the automated pipelines, this should not be a major task.

- **Author reply:** We thank the reviewer for pointing this out. Our PyLipID analyses were calculated over the full trajectories. We apologise for this oversight and have corrected the typographical error of 9.8 μ s in the Figure 2 caption which the reviewer refers to.
- Nonetheless, this remains an important point which we have addressed with further control CG-MD simulations for the SCTR receptor, with three independent repeats initiated from different lipid arrangements. These simulations were used to assess the impact of starting configurations on lipid binding patterns. The results showed consistent lipid occupancy across repeats and between the 10 μ s and 100 μ s simulations, suggesting that the sampling is sufficient and that the key interactions are not strongly biased by the initial set up. Additionally, we calculated other lipid properties, including the diffusion coefficient, for both

the 10 μ s and 100 μ s datasets, to provide additional evidence (new **SI Fig.5-8, and Page 7 lines 146-154**). We agree with the reviewer that this is an important point, and provide further discussion of sampling in both cg and atMD simulations (**Page 26 lines 502-510**). We again note that our trajectories are freely available so PyLipid can be utilised by any interested reader who may wish to test alternative settings.

-The coarse-graining naturally limits Martini's ability to describe conformational entropy (which is an important parameter in lipid-protein interactions, notably with polyunsaturated fatty acids), which is then incorporated into the LJ interactions. Martini also has a limited ability to describe electrostatic interactions. These factors could significantly affect the binding of PIP₂ with polyunsaturated fatty acid chain and a highly-charged headgroup. To this end, I would expect the authors to validate (at least some) of the results involving PIP₂ with atomistic models by fine-graining suitable PIP₂-bound structures observed in coarse-grained simulations.

- **Author reply:** We thank the reviewer for raising these points. By comparison with experiment, the Martini model has shown remarkably strong predictive power when it comes to identifying phosphoinositide interaction sites. We refer the reviewer to a recent review of simulations of phosphoinositides simulations by our coauthor Dr Hedger (PMID: 39793883), and have included a reference to the same in the updated manuscript. Particularly notable examples include the de novo identification of PIP₂ binding sites on Kir channels in 2009 by Stansfeld et al. (PMID: 19839652) in 2009, later confirmed by an X-ray structure solved by the McKinnon group (PMID: 21874019). Similarly for PC2 ion channels (PMID: 34962393). More recently, we performed a combined simulation / native mass spectrometry study in *Nature* revealing PIP₂ as a modulator of GPCR / G-protein coupling, with the Martini simulations being leveraged to de novo identify PIP₂ binding sites (PMID: 29995853), which were subsequently supported by experimental mutagenesis. Moreover these works were performed with earlier iterations of Martini PIP₂ parameters. In the present work we leverage the recently upgraded phosphoinositide parameters set (PMID: 34962393) which theoretically improves accuracy even further.
- To address this point further, we tested if cgMD resolution PIP₂s are readily distinguishable by including a comparison of PI(3,4)P₂ and PI(4, 5)P₂ for GLP1R and GIPR test cases (**Page 28, lines 547-552** and new **Fig. S20**). Comparison of the occupancy profiles shows that at coarse grain resolution PI(3,4)P₂ and PI(4,5)P₂ are similar, thus we have changed the naming from PI(3,4)P₂ to PIP₂ to reflect the simpler model (as is the convention for Martini simulations).
- To assess the cgMD PIP₂ poses at finer resolution, we backmapped to atomistic resolution and assessed the top PIP₂ binding sites for GIPR in the active and inactive states over 500 ns atMD simulations (**Page 14, lines 285-292**). For PIP₂ we show the final poses after 500 ns (new panels in Fig. 4), in which the binding pose is maintained, with some reorganisation of protein side chains (eg Lys397 interaction is reduced in the active state pose):
- **Additional panels in Fig 4:**

- **From Fig. 1. Interactions of PIP₂ with class B1 GPCRs.** The final position of the PIP₂ lipid following 500 ns atomistic simulation is shown for GIPR in the active (top) and inactive state (bottom)
- We have computed lipid occupancy and presented it as a sequence heatmap for the cgMD vs atomistic datasets (GLP1R, GIPR, and SCTR), with results shown next to each other in Fig. S15 (CG cutoff = 7 Å; AT cutoff = 4 Å). Overall, the atomistic binding sites correspond to cg ones, although the cgMD sequence heatmaps tend to show higher occupancy values compared to the atomistic ones.

Optimally, these validation simulations could involve biased simulations to extract PIP₂ binding affinities.

- **Author reply:** Free energy calculations can indeed be useful tools for investigating lipid protein-interactions. Indeed, alongside the work on Arnarez *et al* (PMID: 23405277), we performed the earliest free energy calculation work on lateral lipid-protein interactions using the Martini model (PMID: 27109430, 27807980, 27786441). Not for lack of trying, we never managed to obtain converged (meaningful) calculations for lipid-protein interactions using any atomistic forcefields. From a convergence standpoint it is hard enough dealing with a small molecule ligand in a canonical protein binding pocket, let alone a larger and more complex lipid with many more degrees of freedom and exposed to a slowly diffusing lipid bilayer. Compared to phospholipids, the challenge is made more feasible for cholesterol (PMID: 30358394) with its relatively rigid tetracyclic structure, however such detailed atomistic free energy calculations are beyond the scope of the current work.

-The authors used Chai-1, as it provides "AF3-like accuracy". I wonder why AF3 was not used for comparison?

- **Author reply:** AF3 webportal is currently limited to a pre-defined set of small molecules and thus not capable of complex prediction. Chai-1 allows any small molecule to be inputted via a SMILES string. We have noted this in the manuscript in **Page 12, lines 226-228**.

-The font sizes in Figures are often way too small. For example, it's impossible to read the labels in Fig. 2.

- **Author reply:** We fully agree and apologise for this oversight. We have revised all main text figures to increase label sizes.

-From the general patterns of lipid interactions, can we learn something about interaction motifs?

For example, if some members of the B1 GPCRs do interact and others don't, does this correlate somehow with their sequences?

- **Author reply:** We thank the reviewer for this interesting question. We have chosen not to focus on these motifs, in part due to a recent analysis finding that “92% of cholesterol molecules on GPCR surfaces reside in predictable locations that lack discernible cholesterol-binding motifs” (PMID: PMC9162085). We have now added this reference to the manuscript. The previous GPCR study by Tieleman group (already referenced in our manuscript, PMID: 32272057) also found that CRAC/CARC motifs do not necessarily align with sites observed in MD simulations.

On this note, we foresee that our dataset could be of value to the machine learning community, which may be able to extract and validate meaningful lipid interaction patterns across B1 GPCRs. In particular, we note that the POPC site we identified for GHRHR was found due to the matching residence time pattern to PAC1R (**new Fig. S1**), highlighting a potential route to identifying novel lipid interaction motifs by these methods, which we note in the manuscript (**Page 23,24 lines 438-441**).

-The difference of "occupancy" and "contacts" in Fig. 2 is not clear from the results section.

- **Author reply:** We agree, and have now moved the contacts plot to **Fig S1**. for clarity of Figure 2.

-In Fig. 2 (or in the SI), could the authors show a negative control, i.e. data for a lipid that doesn't have specific interactions.

- **Author reply:** We have now included POPC as a presumed non-regulatory lipid and included this in new **Fig. S1**. Interestingly, POPC shows high residence time interactions with PAC1R in the active state. Upon investigation, we found this corresponded to a rare deep membrane binding pose of POPC. A recent large-scale GPCR study using POPC membranes at atomistic resolution also highlighted multiple instances of POPC burying between TMHs. The same pattern in the residence time plot was observed for GHRHR active state (to a lesser extent), which allowed us, by comparison, to identify this deep membrane site in GHRHR. This has been added to the manuscript as a new short section (**Page 21,22, lines 402-413, new Fig. S19**), and the Discussion (**Page 23,24 lines 433-441**).

From Fig S1, 19:

- **A) From Fig. S1. Analysis of Class B1 GPCRs with presumed non-regulatory lipid POPC.** Residence time plotted as function of ECD, helix, and loop number. **B) From Fig. S19. Rare deep membrane binding events**

of POPC to PAC1R and GHRHR active states. Left: Volumetric density of POPC PO4 headgroups across 3 x 10 μ s repeats. Right: Timestep series showing headgroup positions of deep binding POPC molecule during one replicate.

-It would also be revealing to see the overall tendencies of the proteins to interact with different lipid classes, visualised by, e.g., a bar graph of normalized contacts of all the lipid classes in the simulation system.

- **Author reply:** We thank the reviewer for the suggestion, and we agree this would be interesting. However, due to differences in receptor properties—such as varying numbers of residues and surface areas—as well as the differing proportions of lipid species in the membrane, we believe that a normalized contact bar graph may not accurately reflect the true tendencies of protein-lipid interactions across different GPCRs. In Figures S1–S5, we have chosen to focus on residue-level interactions with cholesterol, PIP2, and GM3, as these are the lipid species most widely implicated in the modulation of GPCR function, although we have included analysis of POPC as addressed in the point above.

-Is the cholesterol binding site near TM6 a non-annular one, so that cholesterol only fits there in the active conformations? Or is there another justification for the interaction observed mainly in the active conformation?

- **Author reply:** We thank the reviewer for this interesting point – based on the global analyses of occupancy (higher for active) vs residence time (no observable difference) we believe this site does indeed correspond to a non-annular lipid site.

-What are "summed [...] residence times"? I think it would be more useful to provide the typical interaction times as these can be compared to other studies.

- **Author reply:** We used summed residence times due to mean residence times containing transient interactions. We now include the overall mean in **new Fig. S1** for comparison to other studies.

-For cholesterol–W^{4.50} interaction, is there a specific pair of Martini beads in cholesterol and tryptophan that has a particularly strong LJ interaction? Or do tryptophan and cholesterol adapt a specific interaction mode? Or do the neighbouring amino acids (and the protein shape) play a role here?

- **Author reply:** This is an interesting point. In our simulations W4.50 consistently showed one of the highest cholesterol occupancies across the B1 family (Fig. S2), and interactions with the conserved tryptophan in ECL2 were also observed, consistent with cryo-EM findings. We believe this is unlikely to result from a particularly strong Lennard-Jones (LJ) interaction between the specific Martini beads for cholesterol and tryptophan, as other tryptophan residues in the sequence exhibited only modest occupancy. This suggests that the interaction at W4.50 is not primarily due to force field bias, but rather driven by the local environment ie neighbouring residues and the overall protein architecture.

-It could be good to point out that cholesterol accesses the membrane core a lot in Martini simulations containing lipids with (poly-)unsaturated acyl chains of lipids. This is due to the repulsion of cholesterol by the double-bond beads that is required to capture liquid-liquid phase separation of ternary lipid mixtures. In atomistic simulations, this effect is significantly smaller.

- **Author reply:** We thank the reviewer for pointing this out. The recent Martini 3 lipid publication (<https://doi.org/10.1021/acscentsci.5c00755>) and our own lipid density analysis both support this observation. We agree this is an important point and have now included it in Results section alongside the computation of cholesterol core access in cgMD vs atomistic simulations (**Page 13, lines 255-258, new Fig. S14**). Nevertheless, we observed a spontaneous deep membrane binding event in the atomistic simulations where cholesterol independently accessed the same deep membrane site (new panel **Fig 4F**).

-Fig. 3 again has too small fonts, panel B is especially hard to read. Panel C, on the other hand, is hard to grasp. I suggest to use either more contrasting colours than yellow & orange, or better yet, extract volumetric density maps of the occupancies.

- **Author reply:** We have remade Figure 3 to improve readability and note that the Panel C data is also expanded in Fig. S11-13. We have now included Supplementary Movie 1 to allow a more complete visualisation of the top cgMD sites for cholesterol and PIP₂.

-How well does PIP2 binding (Fig. 4A) correlate with protein surface charge?

- **Author reply:** The two sites highlighted in the active and inactive state do indeed correlate with charge, with K/R ladders driving this interaction across the B1 family (Fig 4B, residues highlighted in Fig 4CD). We have made a note of this in the main text (**Page 14, line 290**)

-The text in Fig. 4B is unreadable.

- **Author reply:** We have changed the labelling and expanded panel 4B for readability.

-How much does the elastic network affect the ECD dynamics? Does the degree of mobility correlate with the amount of elastic network springs connecting ECD to the TM bundle?

- **Author reply:** The question regarding the elastic network and ECD dynamics in cgMD simulations has also been addressed in response to Reviewer 1. While this is indeed a relevant point, and something we have now explicitly stated in the manuscript (**Page 17, line 336 and Page 26 lines 495-496**) we believe it would be challenging to directly quantify or compare ECD dynamics based on the number of elastic network springs connecting the ECD to the transmembrane bundle, due to the complexity and variability in structural arrangements across different receptors. We have carried out atomistic simulations to analyse the ECD motions further, which in the timeframes accessible to atMD the ECDs remain closely aligned to their initial pose (**Page 17 lines 336-342, new Fig S17,18**).

-With all replicas starting from the same structure, it is very hard to assess the significance of the effect of GM3 on ECD conformation. I think this requires either validation using all-atom simulations, with coarse-grained biased simulations, or unbiased coarse-grained simulations using fully independent replicas, preferably with varying amounts of GM3 present in the membrane.

- **Author reply:** We have performed additional all-atom simulations using different CG2AT2 backmapped frames (in both 10% and 0% GM3 conditions) to further assess the effect of GM3 on ECD conformation. We found that the ECD deviation in the 500 ns timeframe is limited during atMD simulations, thus much longer time frame atMD simulations would need to be carried out to study this further (beyond the scope of this manuscript). We have noted the limitations of sampling for both CGMD and atomistic simulations (**Page 26 lines 502-510**).

-Does the TR-FRET agree with the simulations? If I got it right, in experiments with GLP1R, activation leads to more interaction of ECD and plasma membrane, whereas in simulations the tilt angle is larger for the active case (ECD stands more upright). Also, Eliglustat has no effect on GIPR, yet in simulations GM3 has a larger effect for GIPR.

- **Author reply:** We thank the reviewer for this important question. We carried out the FRET analyses based on our observation from simulations that the ECD motion appears to be modulated by GM3. The concise response is that cgMD simulations capture some but not all of these properties; we have expanded our results section to compare this more clearly (**Page 19, 20 lines 373-385**) as well as describing the limitations in the Discussion (**Page 26, lines 495-501**), but that our combined experimental and simulation results show state-dependent modulation of ECD motion by GM3.

To expand further:

For GLP-1R: simulations showed limited ECD dynamics in the inactive state under both 0% and 10% GM3 conditions, which agrees with the experimental finding that there is no significant difference between the Control and Eliglustat conditions. In the active state, while experiments suggest a significant reduction in ECD dynamics between the Control Ex4 and Eliglustat Ex4 conditions, our simulations did not show a strong GM3-dependent effect on ECD dynamics.

For GIPR: simulations indicated that the ECD is most dynamic in the inactive state, showing a wide range of bend angles under both 0% and 10% GM3 conditions. This supports the experimental interpretation that the GIPR ECD is already highly mobile in the absence of ligand. However, we did not observe a significant reduction in ECD dynamics upon GM3 depletion in our simulations, as was seen experimentally. In the active state, the GIPR ECD adopted a tilted conformation under both GM3 conditions, with no strong GM3 dependency observed—consistent with experimental findings that no significant difference was observed between Control GIP and Eliglustat GIP conditions.

-If PPM 3.0 was used to resolve the membrane positioning of the GPCRs, why does the workflow in Fig. 1 refer to gmx confms, which seems obsolete in this case? Moreover, are the steps performed in CHARMM-GUI (which to my knowledge cannot be scripted) integrated in the aiida approach? Fig. 1 omits them.

- **Author reply:** We thank the reviewer for pointing this out. Due to the complexity of the full workflow and the number of receptors in this dataset, the membrane orientation step is indeed one of the few steps that could not readily be scripted. We have now removed the gmx confms step from Fig. 1 and have noted in the caption that this step cannot be scripted (**Page 7, lines 137-138**). Nevertheless, others have been working to develop software for this step, eg MemPro by the Stansfeld group (in development). We foresee that the full workflow will be fully automated/scripted in the future.

With regard to CHARMM-GUI, none of the CHARMM-GUI steps were used in this workflow except for the initial CHARMM-GUI PDB Reader step, which was used to model back any missing atoms for martinize2 to work. Given the number of receptors in this dataset, we chose to use the CHARMM-GUI PDB Reader, as it is the most straightforward option and heavily used in the MD community. We have added the CHARMM-GUI PDB Reader step into Fig. 1 for clarity and again noted this step cannot be scripted (**Page 7, lines 137-138**).

-The description of simulation methodology is incomplete. For example, information on the handling of non-bonded interactions is missing.

- **Author reply:** We have added the descriptions of the nonbonded interactions in the Methods section. Additionally, we have uploaded all the mdp files used for this manuscript to the same Zenodo depository.

Response to Reviewers' Comments: [COMMSBIO-25-0310A]

We thank the reviewers for their additional helpful comments and suggestions.

Reviewer #1 (Remarks to the Author):

The authors have adequately addressed my questions and concerns. The manuscript has greatly improved with the the new additional control simulations and analyses. The only aspect that is still missing relates to the lines 341-342 of page 17. They wrote "Accordingly, comparison of GM3 contacts showed similar trends for at MD and cgMD (Fig. S18 B,C)." However, Fig S18 does not contain panels B and C. Thus, it is is not possible to discern whether the GMX3 contacts showed similar trends.

Author reply: We are pleased the reviewer finds that we have addressed all questions and concerns. We apologise for the typographic error on Page 17 and have now amended this to refer correctly to Fig. S18.

Reviewer #2 (Remarks to the Author):

The authors have addressed my comments to an adequate degree, and the manuscript is now of suitable quality for publication.

I still recommend the authors to increase font sizes in all figures. At least in their current (raster) format, the text is often unreadable

Author reply: We are pleased the reviewer finds the corrected manuscript suitable for publication. We have now exported all figures in pdf format to resolve the issue of text quality upon conversion.